# Equivariant Manifold Flows

**Isay Katsman\*, Aaron Lou\*, Derek Lim\*, Qingxuan Jiang\***
Cornell University
{isk22, al968, dl772, qj46}@cornell.edu

**Ser-Nam Lim**
Facebook AI
sernam@gmail.com

**Christopher De Sa**
Cornell University
cdesa@cs.cornell.edu

## Abstract

Tractably modelling distributions over manifolds has long been an important goal in the natural sciences. Recent work has focused on developing general machine learning models to learn such distributions. However, for many applications these distributions must respect manifold symmetries—a trait which most previous models disregard. In this paper, we lay the theoretical foundations for learning symmetry-invariant distributions on arbitrary manifolds via equivariant manifold flows. We demonstrate the utility of our approach by learning quantum field theory-motivated invariant $SU(n)$ densities and by correcting meteor impact dataset bias.

## 1 Introduction

Learning probabilistic models for data has long been the focus of many problems in machine learning and statistics. Though much effort has gone into learning models over Euclidean space [6, 20, 21], less attention has been allocated to learning models over non-Euclidean spaces, despite the fact that many problems require a manifold structure. Density learning over non-Euclidean spaces has applications ranging from quantum field theory in physics [44] to motion estimation in robotics [16] to protein-structure prediction in computational biology [22].

Continuous normalizing flows (CNFs) [6, 21] are powerful generative models for learning structure in complex data due to their tractability and theoretical guarantees. Recent work [29, 30] has extended the framework of continuous normalizing flows to the setting of density learning on Riemannian manifolds. However, for many applications in the natural sciences, this construction is insufficient as it cannot properly model necessary symmetries. For example, such symmetry requirements arise when sampling coupled particle systems in physical chemistry [26] or sampling for use in $SU(n)$[1] lattice gauge theories in theoretical physics [3].

More precisely, these symmetries are invariances with respect to action by an isometry subgroup of the underlying manifold.

Figure 1: An example of a density on $SU(3)$ that is invariant to conjugation by $SU(3)$. The $x$-axis and $y$-axis are the angles $\theta_1$ and $\theta_2$ for eigenvalues $e^{i\theta_1}$ and $e^{i\theta_2}$ of a matrix in $SU(3)$. The axis range is $-\pi$ to $\pi$.

---

\* indicates equal contribution

[1] $SU(n)$ denotes the special unitary group $SU(n) = \{X \in \mathbb{C}^{n \times n} \mid X^*X = I, \ \det(X) = 1\}$.

35th Conference on Neural Information Processing Systems (NeurIPS 2021).

For example, consider the task of learning a density on the sphere that is invariant to rotation around an axis; this is an example of learning an isometry subgroup invariant[2] density. For a less trivial example, note that when learning a flow-based sampler for $SU(n)$ in the context of lattice QFT [3], the learned density must be invariant to conjugation by $SU(n)$ (see Figure 1 for a density on $SU(3)$ that exhibits the requisite symmetry).

One might naturally attempt to work with the quotient of the manifold by the relevant isometry subgroup in order to model the invariance. First, note that this structure is not always a manifold, and additional restrictions are needed on the action to ensure the quotient will have a manifold structure[3]. Assuming the quotient is in fact a manifold, one then asks whether an invariant density may be modelled by learning over this quotient with a general manifold density learning method such as NMODE [29]? Though this seems plausible, it is a problematic approach for several reasons:

1. First, it is often difficult to realize necessary constructs (charts, exponential maps, tangent spaces) on the quotient manifold (e.g. this is the case for $\mathbb{RP}^n$, a quotient of $\mathbb{S}^n$ [28]).

2. Second, even if the above constructs can be realized, the quotient manifold often has a boundary, which precludes the use of a manifold CNF. To illustrate this point, consider the simple case of the sphere invariant to rotation about an axis; the quotient manifold is a closed interval, and a CNF would "flow out" on the boundary.

3. Third, even if the quotient is a manifold without boundary for which we have a clear characterization, it may have a discrete structure that induces artifacts in the learned distribution. This is the case for Boyda et al. [3]: the flow construction over the quotient induces abnormalities in the density.

Motivated by the above drawbacks, we design a manifold continuous normalizing flow on the original manifold that maintains the requisite symmetry invariance. Since vanilla manifold CNFs do not maintain said symmetries, we instead construct *equivariant* manifold flows and show they induce the desired invariance. To construct these flows, we present the first general way of designing equivariant vector fields on manifolds. A summary of our paper's contributions is as follows:

- We present a general framework and the requisite theory for learning equivariant manifold flows: in our setup, the flows can be learned over arbitrary Riemannian manifolds while explicitly incorporating symmetries inherent to the problem. Moreover, we prove that the equivariant flows we construct can universally approximate distributions on closed manifolds.

- We demonstrate the efficacy of our approach by learning gauge invariant densities over $SU(n)$ in the context of quantum field theory. In particular, when applied to the densities in Boyda et al. [3], we adhere more naturally to the target geometry and avoid the unnatural artifacts of the quotient construction.

- We highlight the benefit of incorporating symmetries into manifold flow models by comparing directly against previous general manifold density learning approaches. We show that when a general manifold learning model is not aware of symmetries inherent to the problem, the learned density is of considerably worse quality and violates said symmetries. Prior to our work, there did not exist literature that demonstrated the benefits of incorporating isometry group symmetries for learning flows on manifolds, yet we achieve these benefits, and do so through a novel equivariant vector field construction.

## 2  Related Work

Our work builds directly on pre-existing manifold normalizing flow models and enables them to leverage inherent symmetries through equivariance. In this section we cover important developments from the relevant fields: manifold normalizing flows and equivariant machine learning.

---

[2]This specific isometry subgroup is known as the isotropy group at a point of the sphere intersecting the axis.

[3]In particular, the isometry subgroup action needs to be smooth, free, and proper to ensure the quotient will be a manifold by the Quotient Manifold Theorem [28].

**Normalizing Flows on Manifolds**   Normalizing flows on Euclidean space have long been touted as powerful generative models [6, 10, 21]. Similar to GANs [20] and VAEs [24], normalizing flows learn to map samples from a tractable prior density to a target density. However, unlike the aforementioned models, normalizing flows account for changes in volume, enabling exact evaluation of the output probability density. In a rather concrete sense, this makes them theoretically principled. As such, they are ideal candidates for generalization beyond the Euclidean setting, where a careful, theoretically principled modelling approach is necessary.

Motivated by recent developments in geometric deep learning [4], many methods have extended normalizing flows to Riemannian manifolds. Rezende et al. [38] introduced constructions specific to tori and spheres, while Bose et al. [2] introduced constructions for hyperbolic space. Following this work, Falorsi and Forré [15], Lou et al. [29], Mathieu and Nickel [30] concurrently introduced a general construction by extending Neural ODEs [6] to the setting of Riemannian manifolds. Our work takes inspiration from the methods of Lou et al. [29], Mathieu and Nickel [30] and generalizes them further to enable learning that takes into account symmetries of the target density.

**Equivariant Machine Learning**   Motivated by the observation that many classic neural network architectures incorporate symmetry as an inductive bias, recent work has leveraged symmetries inherent in data through the concept of equivariance [7–9, 18, 27, 37]. Köhler et al. [26], in particular, used equivariant normalizing flows to enable learning symmetric densities over Euclidean space. The authors note their approach is better suited to density learning in some physical chemistry settings (when compared to general purpose normalizing flows), since they take into account the symmetries of the problem.

Symmetries also appear naturally in the context of learning densities over manifolds. While in many cases symmetry can be a good inductive bias for learning[4], for certain test tasks it is a strict requirement. For example, Boyda et al. [3] introduced equivariant flows on $SU(n)$ for use in lattice gauge theories, where the modelled distribution must be conjugation invariant. However, beyond conjugation invariant learning on $SU(n)$ [3], not much other work has been done for learning invariant distributions over manifolds. Our work bridges this gap by introducing the first general equivariant manifold normalizing flow model for arbitrary manifolds and symmetries.

## 3   Background

In this section, we provide a terse overview of necessary concepts for understanding our paper. In particular, we address fundamental notions from Riemannian geometry as well as the basic set-up of normalizing flows on manifolds. For a more detailed introduction to Riemannian geometry, we refer the reader to textbooks such as Lee [28] and Kobyzev et al. [25].

### 3.1   Riemannian Geometry

A Riemannian manifold $(\mathcal{M}, h)$ is an $n$-dimensional manifold with a smooth collection of inner products $(h_x)_{x \in \mathcal{M}}$ for every tangent space $T_x \mathcal{M}$. The Riemannian metric $h$ induces a distance $d_h$ on the manifold.

A diffeomorphism $f : \mathcal{M} \to \mathcal{M}$ is a differentiable bijection with differentiable inverse. A diffeomorphism $f : \mathcal{M} \to \mathcal{M}$ is called an isometry if $h(D_x f(u), D_x f(v)) = h(u, v)$ for all tangent vectors $u, v \in T_x \mathcal{M}$ where $D_x f$ is the differential of $f$. Note that isometries preserve the manifold distance function. The collection of all isometries forms a group $G$, which we call the isometry group of the manifold $\mathcal{M}$.

Riemannian metrics also allow for a natural analogue of gradients on $\mathbb{R}^n$. For a function $f : \mathcal{M} \to \mathbb{R}$, we define the Riemannian gradient $\nabla_x f$ to be the vector on $T_x \mathcal{M}$ such that $h(\nabla_x f, v) = D_x f(v)$ for $v \in T_x \mathcal{M}$.

### 3.2   Normalizing Flows on Manifolds

**Manifold Normalizing Flow**   Let $(\mathcal{M}, h)$ be a Riemannian manifold. A normalizing flow on $\mathcal{M}$ is a diffeomorphism $f_\theta : \mathcal{M} \to \mathcal{M}$ (parametrized by $\theta$) that transforms a prior density $\rho$ to model

---

[4]For example, asteroid impacts on the sphere can be modelled as being approximately invariant to rotation about the Earth's axis.

density $\rho_{f_\theta}$. The model distribution can be computed via the Riemannian change of variables[5]:

$$\rho_{f_\theta}(x) = \rho\left(f_\theta^{-1}(x)\right)\left|\det_h D_x f_\theta^{-1}\right|.$$

**Manifold Continuous Normalizing Flow**   A manifold continuous normalizing flow with base point $z$ is a function $\gamma : [0, \infty) \to \mathcal{M}$ that satisfies the manifold ODE

$$\frac{d\gamma(t)}{dt} = X(\gamma(t), t), \qquad \gamma(0) = z.$$

We define $F_{X,T} : \mathcal{M} \to \mathcal{M}, z \mapsto F_{X,T}(z)$ to map any base point $z \in \mathcal{M}$ to the value of the CNF starting at $z$, evaluated at time $T$. This function is known as the (vector field) flow of $X$.

### 3.3   Equivariance and Invariance

Let $G$ be an isometry subgroup of $\mathcal{M}$. We notate the action of an element $g \in G$ on $\mathcal{M}$ by the map $L_g : \mathcal{M} \to \mathcal{M}$.

**Equivariant and Invariant Functions**   We say that a function $f : \mathcal{M} \to \mathcal{N}$ is equivariant if, for all isometries $g_x : \mathcal{M} \to \mathcal{M}$ and $g_y : \mathcal{N} \to \mathcal{N}$, we have $f \circ g_x = g_y \circ f$. We say a function $f : \mathcal{M} \to \mathcal{N}$ is invariant if $f \circ g_x = f$.

**Equivariant Vector Fields**   Let $X : \mathcal{M} \times [0, \infty) \to T\mathcal{M}, X(m, t) \in T_m\mathcal{M}$ be a time-dependent vector field on manifold $\mathcal{M}$, with base point $x_0 \in \mathcal{M}$. $X$ is a $G$-equivariant vector field if $\forall (m, t) \in \mathcal{M} \times [0, \infty), X(L_g m, t) = (D_m L_g)X(m, t)$.

**Equivariant Flows**   A flow $f : \mathcal{M} \to \mathcal{M}$ is $G$-equivariant if it commutes with actions from $G$, i.e. we have $L_g \circ f = f \circ L_g$.

**Invariance of Density**   A density $\rho$ on a manifold $\mathcal{M}$ is $G$-invariant if, for all $g \in G$ and $x \in \mathcal{M}$, $\rho(L_g x) = \rho(x)$, where $L_g$ is the action of $g$ on $x$.

## 4   Invariant Densities from Equivariant Flows

Our goal in this section is to describe a tractable way to learn a density over a manifold that obeys a symmetry given by an isometry subgroup $G$. Since this cannot be done directly and it is not clear how a manifold continuous normalizing flow can be altered to preserve symmetry, we will derive the following implications to yield a tractable solution:

1. $G$**-invariant potential** $\Rightarrow$ $G$**-equivariant vector field (Theorem 1).** We show that given a $G$-invariant potential function $\Phi : \mathcal{M} \to \mathbb{R}$, the vector field $\nabla\Phi$ is $G$-equivariant.

2. $G$**-equivariant vector field** $\Rightarrow$ $G$**-equivariant flow (Theorem 2).** We show that a $G$-equivariant vector field on $\mathcal{M}$ uniquely induces a $G$-equivariant flow.

3. $G$**-equivariant flow** $\Rightarrow$ $G$**-invariant density (Theorem 3).** We show that given a $G$-invariant prior $\rho$ and a $G$-equivariant flow $f$, the flow density $\rho_f$ is $G$-invariant.

These are constructed in the same spirit as the theorems in Köhler et al. [26] (which also appeared in Papamakarios et al. [34]), although we note that our results are significantly more general. In addition to extending the domain to Riemannian manifolds, we consider arbitrary symmetry groups while Köhler et al. [26] only considers the linear Lie group $SO(n)$. As a result, our proof techniques are based on heavy geometric machinery instead of straightforward linear algebra techniques.

If we have a prior distribution on the manifold that obeys the requisite invariance, then the above implications show that we can use a $G$-invariant potential to produce a flow that, in tandem with the CNF framework, learns an output density with the desired invariance. We claim that constructing a $G$-invariant potential function on a manifold is far simpler than directly parameterizing a $G$-invariant density or a $G$-equivariant flow. We shall give explicit examples of $G$-invariant potential constructions in Section 5.2 that induce a desired density invariance.

---

[5]Here, $\det_h$ is the determinant function with volume induced by the Riemannian metric $h$.

Moreover, we show in Theorem 4 that considering equivariant flows generated from invariant potential functions suffices to learn any smooth distribution over a closed manifold, as measured by Kullback-Leibler divergence.

We defer the proofs of all theorems to the appendix.

## 4.1 Equivariant Gradient of Potential Function

We start by showing how to construct $G$-equivariant vector fields from $G$-invariant potential functions.

To design an equivariant vector field $X$, it is sufficient to set the vector field dynamics of $X$ as the gradient of some $G$-invariant potential function $\Phi : \mathcal{M} \to \mathbb{R}$. This is formalized in the following theorem.

**Theorem 1.** *Let $(\mathcal{M}, h)$ be a Riemannian manifold and $G$ be its group of isometries (or an isometry subgroup). If $\Phi : \mathcal{M} \to \mathbb{R}$ is a smooth $G$-invariant function, then the following diagram commutes for any $g \in G$:*

$$
\begin{array}{ccc}
\mathcal{M} & \xrightarrow{L_g} & \mathcal{M} \\
\downarrow{\scriptstyle \nabla\Phi} & & \downarrow{\scriptstyle \nabla\Phi} \\
T\mathcal{M} & \xrightarrow{DL_g} & T\mathcal{M}
\end{array}
$$

*or $\nabla_{L_g u}\Phi = D_u L_g(\nabla_u \Phi)$. Hence $\nabla\Phi$ is a $G$-equivariant vector field. This condition is also tight in the sense that it only occurs if $G$ is the isometry subgroup.*

Hence, as long as one can construct a $G$-invariant potential function, one can obtain the desired equivariant vector field. By this construction, a parameterization of $G$-invariant potential functions yields a parameterization of (some) $G$-equivariant vector fields.

## 4.2 Constructing Equivariant Manifold Flows from Equivariant Vector Fields

To construct equivariant manifold flows, we will use tools from the theory of manifold ODEs. In particular, there exists a natural correspondence between equivariant flows and equivariant vector fields. We formalize this in the following theorem:

**Theorem 2.** *Let $(\mathcal{M}, h)$ be a Riemannian manifold, and $G$ be its isometry group (or one of its subgroups). Let $X$ be any time-dependent vector field on $\mathcal{M}$, and $F_{X,T}$ be the flow of $X$. Then $X$ is a $G$-equivariant vector field if and only if $F_{X,T}$ is a $G$-equivariant vector field flow.*

Hence we can obtain an equivariant flow from an equivariant vector field, and vice versa.

## 4.3 Invariant Manifold Densities from Equivariant Flows

We now show that $G$-equivariant flows induce $G$-invariant densities. Note that we require the group $G$ to be an isometry subgroup in order to control the density of $\rho_f$, and the following theorem does not hold for general diffeomorphism subgroups.

**Theorem 3.** *Let $(\mathcal{M}, h)$ be a Riemannian manifold, and $G$ be its isometry group (or one of its subgroups). If $\rho$ is a $G$-invariant density on $\mathcal{M}$, and $f$ is a $G$-equivariant diffeomorphism, then $\rho_f$ is also $G$-invariant.*

In the context of manifold normalizing flows, Theorem 3 implies that if the prior density on $\mathcal{M}$ is $G$-invariant and the flow is $G$-equivariant, the resulting output density will be $G$-invariant. In the context of the overall set-up, this reduces the problem of constructing a $G$-invariant density to the problem of constructing a $G$-invariant potential function.

## 4.4 Sufficiency of Flows Generated via Invariant Potentials

It is unclear whether equivariant flows induced by invariant potentials can learn arbitrary invariant distributions over manifolds. In particular, it is reasonable to have some concerns about limited expressivity, since it is unclear whether any equivariant flow can be generated in this way. We alleviate

these concerns for our use cases by proving that equivariant flows obtained from invariant potential functions suffice to learn any smooth invariant distribution over a closed manifold, as measured by Kullback-Leibler (KL) divergence.

**Theorem 4.** *Let $(\mathcal{M}, h)$ be a closed Riemannian manifold. Let $\pi$ be a smooth, non-vanishing distribution over $\mathcal{M}$, which will act as our target distribution. Let $\rho_t$ be a distribution over said manifold parameterized by a real time variable $t$, with $\rho_0$ acting as the initial distribution. Let $D_{KL}(\rho_t \| \pi)$ denote the Kullback–Leibler divergence between distributions $\rho_t$ and $\pi$. If we choose a $g : \mathcal{M} \to \mathbb{R}$ such that*

$$g(x) = \log \left( \frac{\pi(x)}{\rho_t(x)} \right),$$

*and if $\rho_t$ evolves with $t$ as the distribution of a flow according to $g$, it follows that*

$$\frac{\partial}{\partial t} D_{KL}(\rho_t \| \pi) = - \int_{\mathcal{M}} \rho_t \exp(g) \|\nabla g\|^2 \, dx = - \int_{\mathcal{M}} \pi \|\nabla g\|^2 \, dx$$

*implying convergence of $\rho_t$ to $\pi$ in $KL$. Moreover, the exact diffeomorphism that takes us from $\rho_0 \to \pi$ is as follows. Given some initial point $x \in \mathcal{M}$, let $u(t)$ be the solution to the initial value problem given by:*

$$\frac{du(t)}{dt} = \nabla g(t), \qquad u(0) = x$$

*The desired diffeomorphism maps $x$ to $\lim_{t \to \infty} u(t)$.*

Hence if the target distribution is $\pi$, the current distribution is $\rho_0$, and $g$ as defined above is the potential from which the flow controlling the evolution of $\rho_t$ is obtained, then $\rho_t$ converges to $\pi$ in $KL$. This means that considering flows generated by invariant potential functions is sufficient to learn any smooth invariant target distribution on a closed manifold (as measured by KL divergence).

## 5 Learning Invariant Densities with Equivariant Flows

In this section, we discuss implementation details of the methodology given in Section 4. In particular, we describe the equivariant manifold flow model, provide two examples of invariant potential constructions on different manifolds, and discuss how training is performed depending on the target task.

### 5.1 Equivariant Manifold Flow Model

For our equivariant flow model, we first construct a $G$-invariant potential function $\Phi : \mathcal{M} \to \mathbb{R}$ (we show how to construct these potentials in Section 5.2). The equivariant flow model works by using automatic differentiation [35] on $\Phi$ to obtain $\nabla \Phi$, using this $\nabla \Phi$ for the vector field, and integrating in a step-wise fashion over the manifold. Specifically, forward integration and change-in-density (divergence) computations utilize the Riemannian Continuous Normalizing Flows [30] framework. This flow model is used in tandem with a specific training procedure (described in Section 5.3) to obtain a $G$-invariant model density that approximates some target.

### 5.2 Constructing $G$-invariant Potential Functions

In this subsection, we present two constructions of invariant potentials on manifolds. Note that a symmetry of a manifold (i.e. action by an isometry subgroup) will leave part of the manifold free. The core idea of our invariant potential construction is to parameterize a neural network on the free portion of the manifold. While the two constructions we give below are certainly not exhaustive, they illustrate the versatility of our method, which is applicable to general manifolds and symmetries.

### 5.2.1 Isotropy Invariance on $S^2$

Consider the sphere $S^2$, which is the Riemannian manifold $\{v \in \mathbb{R}^3 : \|v\| = 1\}$ with the induced pullback metric. The isotropy group for a point $v$ is defined as the subgroup of the isometry group which fixes $v$, i.e. the set of rotations around an axis that passes through $v$. In practice, we let $v = (0, 0, 1)$, so the isotropy group is the group of rotations on the $xy$-plane. An isotropy invariant density would be invariant to such rotations, and hence would look like a horizontally-striped density on the sphere (see Figure 4a).

**Invariant Potential Parameterization**   We design an invariant potential by applying a neural network to the free parameter. In the case of our specific isotropy group listed above, the free parameter is the $z$-coordinate. The invariant potential is simply a 2-input neural network with the spatial input being the $z$-coordinate and the time input being the time during integration. As a result of this design, we see that the only variance in the learned distribution that uses this potential will be along the $z$-axis, as desired.

**Prior Distributions**   For proper learning with a normalizing flow, we need a prior distribution on the sphere that respects the isotropy invariance. There are many isotropy invariant potentials on the sphere. Natural choices include the uniform density (which is invariant to all rotations) and the wrapped distribution with the center at $v$ [33, 40]. For our experiments, we use the uniform density.

### 5.2.2   Conjugation Invariance on $SU(n)$

For many applications in physics (specifically gauge theory and lattice quantum field theory), one works with the Lie Group $SU(n)$ — the group of unitary matrices with determinant 1. In particular, when modelling probability distributions on $SU(n)$ for lattice QFT, the desired distribution must be invariant under conjugation by $SU(n)$ [3]. Conjugation is an isometry on $SU(n)$ (see Appendix A.5), so we can model probability distributions invariant under this action with our developed theory.

**Invariant Potential Parameterization**   We want to construct a conjugation invariant potential function $\Phi : SU(n) \to \mathbb{R}$. Note that matrix conjugation preserves eigenvalues. Thus, for a function $\Phi : SU(n) \to \mathbb{R}$ to be invariant to matrix conjugation, it has to act on the eigenvalues of $x \in SU(n)$ as a multi-set.

We can parameterize such potential functions $\Phi$ by the DeepSet network from [45]. DeepSet is a permutation invariant neural network that acts on the eigenvalues, so the mapping of $x \in SU(n)$ is $\Phi(x) = \hat{\Phi}(\{\lambda_1(x), \ldots, \lambda_n(x)\})$ for some set function $\hat{\Phi}$. We append the integration time to the input of the standard neural network layers in the DeepSet network.

As a result of this design, we see that the only variance in the learned distribution will be amongst non-similar matrices, while all similar matrices will be assigned the same density value.

**Prior Distributions**   For the prior distribution of the flow, we need a distribution that respects the matrix conjugation invariance. We use the Haar measure on $SU(n)$, which is the uniform density over this manifold that is symmetric under gauge symmetry [3]. The volume element of the Haar measure is given for an $x \in SU(n)$ as $\text{Haar}(x) = \prod_{i<j} |\lambda_i(x) - \lambda_j(x)|^2$. We can sample from and compute the log probabilities with respect to this distribution efficiently with standard matrix computations [32].

### 5.3   Training Paradigms for Equivariant Manifold Flows

There are two notable ways in which we can use the model described in Section 5.1. Namely, we can use it to learn to sample from a distribution for which we have a density function, or we can use it to learn the density given a way to sample from the distribution. These training paradigms are useful in different contexts, as we will see in Section 6.

**Learning to sample given an exact density.**   In certain settings, we are given an exact density and the task is to learn a tractable sampler for the distribution. For example in Boyda et al. [3], we are given conjugation-invariant densities on $SU(n)$ for which we know the exact density function (without knowledge of any normalizing constants). In contrast to procedures for normalizing flow training that use negative log-likelihood based losses, we do not have access to samples from the target distribution. Instead, we train our models by sampling from the Haar distribution on $SU(n)$, computing the KL divergence between the probabilities that our model assigns to these samples and the probabilities of the target distribution evaluated at these samples, and backpropagating from this KL divergence loss. When this loss is minimized, we can sample from the target distribution by sampling the prior, then forwarding the prior samples through our model. In the context of Boyda et al. [3], such a flow-based sampler is important for modelling gauge theories.

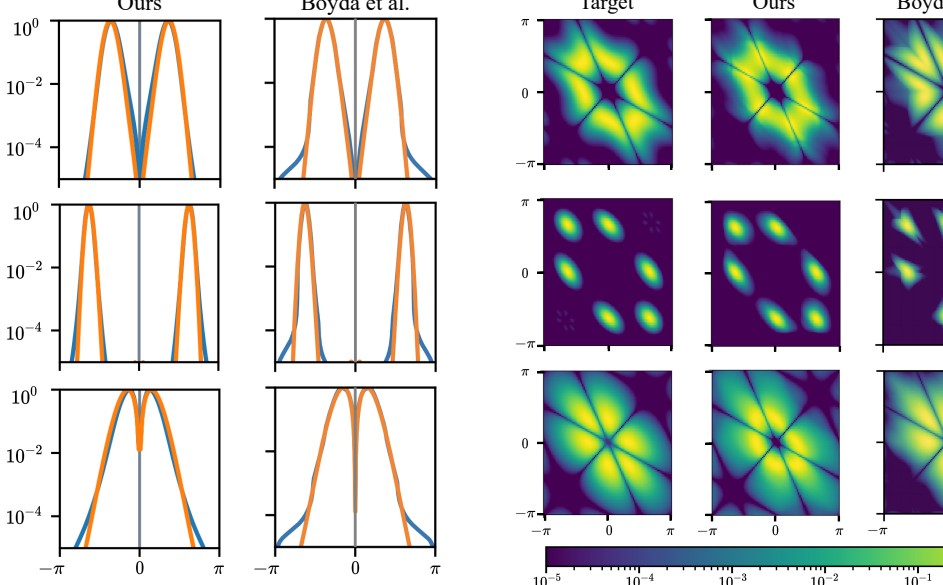

(a) $SU(2)$ learned densities from (Left) our model and (Right) the Boyda et al. [3] model. The target densities are in orange, while the model densities are in blue. The $x$-axis is $\theta$ for an eigenvalue $e^{i\theta}$ of a matrix in $SU(2)$ (note the second eigenvalue is determined as $e^{-i\theta}$). Our model has much better behavior in low-density regions (Boyda et al. [3] fails to eliminate mass around $\pm\pi$) and more smoothly captures the targets in high-density regions.

(b) $SU(3)$ learned densities from (Middle) our model and (Right) the Boyda et al. [3] model for different target densities (Left). The $x$-axis and $y$-axis are the angles $\theta_1$ and $\theta_2$ for eigenvalues $e^{i\theta_1}$ and $e^{i\theta_2}$ of a matrix in $SU(3)$ (note the third eigenvalue is determined as $e^{-i\theta_1-i\theta_2}$), and the probabilities correspond to colors on a logarithmic scale. Our model better captures the geometry of the target densities and does not exhibit the discrete artifacts of the Boyda et al. [3] model.

Figure 2: Comparison of learned densities on (a) $SU(2)$ and (b) $SU(3)$. All densities are normalized to have maximum value 1.

**Learning the density given a sampler.** In other settings, we are given a way to sample from a target distribution and want to learn the precise density for downstream tasks. For this setting, we sample the target distribution, use our flow to map it to a tractable prior, and use a negative log-likelihood-based loss. The flow will eventually learn to assign higher probabilities in sampled regions, and in doing so, will learn to approximate the target density.

## 6   Experiments

In this section, we utilize instantiations of equivariant manifold flows to learn densities over various manifolds of interest that are invariant to certain symmetries. First, we construct flows on $SU(n)$ that are invariant to conjugation by $SU(n)$; these are useful for lattice quantum field theory [3]. In this setting, our model outperforms the construction of Boyda et al. [3].

As a second application, we model asteroid impacts on Earth by constructing flow models on $S^2$ that are invariant to the isotropy group that fixes the north pole. Our approach is able to overcome dataset bias, as only land impacts are reported in the dataset.

Finally, to demonstrate the need for enforcing equivariance of flow models, we directly compare our flow construction with a general purpose flow while learning a density with an inherent symmetry. The densities we decided to use for this purpose are sphere densities that are invariant to action by the isotropy group. Our model is able to learn these densities much better than previous manifold ODE models that do not enforce equivariance of flows [29], thus showing the ability of our model to leverage the desired symmetries. In fact, even on simple isotropy-invariant densities, our model succeeds while the free model without equivariance fails.

## 6.1 $SU(n)$ **Gauge Equivariant Neural Network Flows**

Learning $SU(n)$ gauge equivariant neural network flows is important for obtaining good flow-based samplers of densities on $SU(n)$ useful for lattice quantum field theory [3]. We compare our model for $SU(n)$ gauge equivariant flows (Section 5.2.2) with that of Boyda et al. [3]. For the sake of staying true to the application area, we follow the framework of Boyda et al. [3] in learning densities on $SU(n)$ that are invariant to conjugation by $SU(n)$. In particular, our goal is to learn a flow to model a target distribution so that we may efficiently sample from it.

As mentioned above in Section 5.3, this setting follows the first paradigm in which we are given exact density functions and learn how to sample.

For the actual architecture of our equivariant manifold flows, we parameterize our potentials as DeepSet networks on eigenvalues as detailed in Section 5.2.2. The prior distribution for our model is also the Haar (uniform) distribution on $SU(n)$. Further training details are given in Appendix C.1.

### 6.1.1 $SU(2)$

Figure 2a displays learned densities for our model and the model of Boyda et al. [3] in the case of three particular densities on $SU(2)$ described in Appendix C.2.1. While both models match the target distributions well in high-density regions, we find that our model exhibits a considerable improvement in lower-density regions, where the tails of our learned distribution decay faster. By contrast, the model of Boyda et al. [3] seems to be unable to reduce mass near $\pm\pi$, a possible consequence of their construction. Even in high-density regions, our model appears to vary smoothly, with fewer unnecessary bumps and curves when compared to the densities of the model in Boyda et al. [3].

### 6.1.2 $SU(3)$

Figure 2b displays learned densities for our model and the model of Boyda et al. [3] in the case of three particular densities on $SU(3)$ described in Appendix C.2.2. In this case, we see that our models fit the target densities more accurately and better respect the geometry of the target distribution. Indeed, while the learned densities of Boyda et al. [3] are often sharp and have pointed corners, our models learn densities that vary smoothly and curve in ways that are representative of the target distributions.

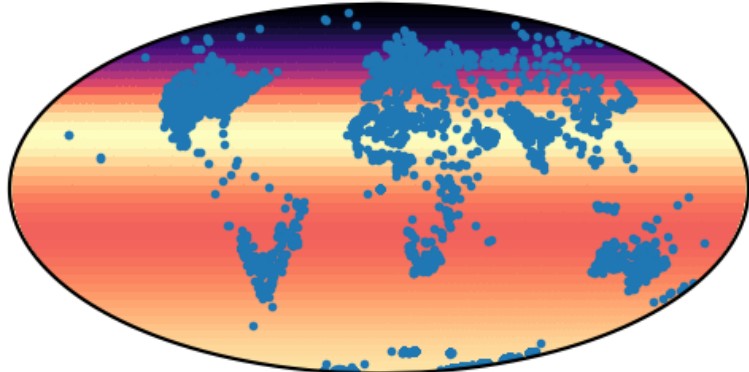

Figure 3: Our modelled distribution of meteor impacts in Meteorite Landings [31]. The true impacts are marked in blue and our isotropy invariant density is shown in the background. Note that a regular manifold normalizing flow would instead model impacts only on land as the dataset does not include any ocean impacts.

## 6.2 Asteroid Impact Dataset Bias Correction

We also showcase our model's ability to correct for dataset bias. In particular, we consider the test case of modelling asteroid impacts on Earth. Towards this end, many preexisting works have compiled locations of previous asteroid impacts [14, 31], but modelling these datasets is challenging since they are inherently biased. In particular, all recorded impacts are found on land. However,

ocean impacts are also dangerous [42] and should be properly modelled. To correct for this bias, we note that the distribution of asteroid impacts should be invariant with respect to the rotation of the Earth. We apply our isotropy invariant $S^2$ flow (described in Section 5.2.1) to model the asteroid impact locations given by the dataset Meteorite Landings [31] [6]. Training happens in the setting of the second paradigm described in Section 5.3, since we can easily sample the target distribution and aim to learn the density. We visualize our results in Figure 3.

## 6.3 Modelling Invariance Matters

We also show that our equivariant condition on the manifold flow matters for efficient and accurate training when the target distribution is invariant. In particular, we again consider the sphere under the action of the isotropy group. We try to learn the isotropy invariant density given in Figure 4a and compare the results of our equivariant flow against those of a predefined manifold flow that does not explicitly model the symmetry [29]. While other manifold flow models have been proposed for the sphere [38], NMODE outperforms them [29], so we use NMODE as a strong baseline. We train for 100 epochs with a learning rate of 0.001 and a batch size of 200; our results are shown in Figure 4.

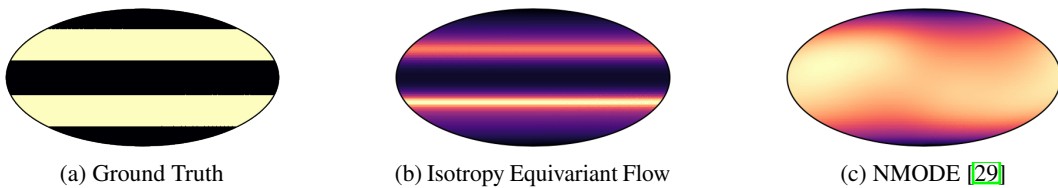

(a) Ground Truth    (b) Isotropy Equivariant Flow    (c) NMODE [29]

Figure 4: We compare the equivariant manifold flow and a regular manifold flow (implemented with NMODE [29]) on an invariant dataset. Note that our model is able to accurately capture the ground truth data distribution while NMODE struggles.

Despite our equivariant flow having fewer parameters (as both flows have the same width and the equivariant flow has an input dimension of 1), our model is able to capture the distribution much better than NMODE [29]. This is due to the inductive bias of our equivariant model which explicitly leverages the underlying symmetry.

## 7 Conclusion

In this work, we introduce equivariant manifold flows in a fully general context and provide the necessary theory to ensure our construction is principled. We also demonstrate the efficacy of our approach in the context of learning conjugation invariant densities over $SU(2)$ and $SU(3)$, which is an important task for sampling $SU(n)$ lattice gauge theories in quantum field theory. In particular, we show that our method can more naturally adhere to the geometry of the target densities when compared to prior work while being more generally applicable. We also present an application to modelling asteroid impacts and demonstrate the necessity of modelling existing invariances by comparing against a regular manifold flow.

**Further considerations.** While our theory and implementations have utility in very general settings, there are still some limitations that could be addressed in future work. Further research may focus on finding other ways to generate equivariant manifold flows that do not rely on the construction of an invariant potential, and perhaps additionally on showing that such methods are sufficiently expressive to learn over open manifolds. Our models also require a fair bit of tuning to achieve results as strong as we demonstrate. Finally, we note that our theory and learning algorithm are too abstract for us to be sure of the future societal impacts. Still, we advance the field of deep generative models, which is known to have potential for negative impacts through malicious generation of fake images and text. Nevertheless, we do not expect this work to have negative effects in this area, as our applications are not in this domain.

---

[6]This dataset was released by NASA without a specified license.

## Acknowledgements

We would like to thank Facebook AI for funding equipment that made this work possible. In addition, we thank the National Science Foundation for awarding Prof. Christopher de Sa a grant that helps fund this research effort (NSF IIS-2008102) and for supporting Aaron Lou with a graduate student fellowship. We would also like to acknowledge Jonas Köhler and Denis Boyda for their useful insights.

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
