# Appendix

## A  Proof of Theorems

In this section, we restate and prove the theorems in Section 4. These give the theoretical foundations that we use to build our models. Prior work [17, 43] addresses some of the results we formalize below.

### A.1  Proof of Theorem 1

**Theorem 1.** *Let $(\mathcal{M}, h)$ be a Riemannian manifold and $G$ be its group of isometries (or an isometry subgroup). If $\Phi : \mathcal{M} \to \mathbb{R}$ is a smooth $G$-invariant function, then the following diagram commutes for any $g \in G$:*

$$
\begin{array}{ccc}
\mathcal{M} & \xrightarrow{\ L_g\ } & \mathcal{M} \\
\downarrow{\scriptstyle \nabla\Phi} & & \downarrow{\scriptstyle \nabla\Phi} \\
T\mathcal{M} & \xrightarrow{\ DL_g\ } & T\mathcal{M}
\end{array}
$$

*or $\nabla_{L_g u}\Phi = D_u L_g(\nabla_u \Phi)$. This is condition is also tight in the sense that it only occurs if $G$ is the group of isometries.*

*Proof.* We first recall the Riemannian gradient chain rule:

$$\nabla_u(\Phi \circ L_g) = (D_u L_g)^\top (\nabla_{L_g u}\Phi)$$

where $(D_u L_g)^\top : T_{L_g u}\mathcal{M} \to T_u \mathcal{M}$ is the "adjoint" given by

$$h\left(D_u L_g(v), w\right) = h\left(v, (D_u L_g)^\top(w)\right).$$

Since $L_g$ is an isometry, we also have

$$h(x, y) = h\left(D_u L_g(x), D_u L_g(y)\right).$$

Combining the above two equations gives

$$h(x, y) = h(D_u L_g(x), D_u L_g(y)) = h\left(x, (D_u L_g)^\top\left(D_u L_g(y)\right)\right),$$

which implies for all $y$,

$$h\left(x, y - (D_u L_g)^\top(D_u L_g(y))\right) = 0.$$

Since $h$ is a Riemannian metric (even pseudo-metric works due to non-degeneracy), we must have that $(D_u L_g)^\top \circ (D_u L_g) = I$.

To complete the proof, we recall that $\Phi = \Phi \circ L_g$, and this combined with chain rule gives

$$\nabla_u \Phi = \nabla_u(\Phi \circ L_g) = (D_u L_g)^\top(\nabla_{L_g u}\Phi).$$

Now applying $D_u L_g$ on both sides gives

$$\nabla_{L_g u}\Phi = D_u L_g \nabla_u \Phi$$

which is exactly what we want to show.

We see that this is an "only if" condition because we must necessarily get that the adjoint is the inverse, which implies that $L_g$ is an isometry. $\qquad\square$

## A.2 Proof of Theorem 2

**Theorem 2.** *Let $(\mathcal{M}, h)$ be a Riemannian manifold, and $G$ be its isometry group (or one of its subgroups). Let $X$ be any time-dependent vector field on $\mathcal{M}$, and $F_{X,T}$ be the flow of $X$. Then $X$ is an $G$-equivariant vector field if and only if $F_{X,T}$ is a $G$-equivariant flow for any $T \in [0, +\infty)$.*

*Proof.* **$G$-equivariant $X \Rightarrow G$-equivariant $F_{X,T}$.** We invoke the following lemma from Lee [28, Corollary 9.14]:

**Lemma 1.** *Let $F : \mathcal{M} \to \mathcal{N}$ be a diffeomorphism. If $X$ is a smooth vector field over $\mathcal{M}$ and $\theta$ is the flow of $X$, then the flow of $F_* X$ ($F_*$ is another notation for the differential of $F$) is $\eta_t = F \circ \theta_t \circ F^{-1}$, with domain $N_t = F(M_t)$ for each $t \in \mathbb{R}$.*

Examine $L_g$ and its action on $X$. Since $X$ is $G$-equivariant, we have for any $(x, t) \in \mathcal{M} \times [0, +\infty)$,

$$((L_g)_* X)(x, t) = (D_{L_g^{-1}(x)} L_g) X(L_g^{-1}(x), t) = X(L_g \circ L_g^{-1}(x), t) = X(x, t)$$

so it follows that $(L_g)_* X = X$. Applying the lemma above, we get:

$$F_{(L_g)_* X, T} = L_g \circ F_{X,T} \circ L_g^{-1}$$

and, by simplifying, we get that $F_{X,T} \circ L_g = L_g \circ F_{X,T}$, as desired.

**$G$-equivariant $X \Leftarrow G$-equivariant $F_{X,T}$.** This direction follows from the chain rule. If $F_{X,T}$ is $G$-equivariant, then at all times we have:

$$
\begin{aligned}
(D_m L_g)\left(X(F_{X,t}(m), t)\right) = (D_m L_g)\left(\frac{d}{dt} F_{X,T}(m)\right) \qquad &\text{(definition)} \\
= \frac{d}{dt}(L_g \circ F_{X,T})(m) \qquad &\text{(chain rule)} \\
= \frac{d}{dt} F_{X,T}(L_g m) \qquad &\text{(equivariance)} \\
= X(L_g(F_{X,t}(m)), t) \qquad &\text{(definition)}
\end{aligned}
$$

This concludes the proof of the backward direction. $\qquad\square$

## A.3 Proof of Theorem 3

**Theorem 3.** *Let $(\mathcal{M}, h)$ be a Riemannian manifold, and $G$ be its isometry group (or one of its subgroups). If $\rho$ is a $G$-invariant density on $\mathcal{M}$, and $f$ is a $G$-equivariant diffeomorphism, then $\rho_f(x)$ is also $G$-invariant.*

*Proof.* We wish to show $\rho_f(x)$ is also $G$-invariant, i.e. $\rho_f(L_g x) = \rho_f(x)$ for all $g \in G, x \in \mathcal{M}$. We first recall the definition of $\rho_f$:

$$\rho_f(x) = \rho\left(f^{-1}(x)\right)\left|\det \frac{\partial f^{-1}(x)}{\partial x}\right| = \rho\left(f^{-1}(x)\right)\left|\det J_{f^{-1}}(x)\right|.$$

Since $f \in C^1(\mathcal{M}, \mathcal{M})$ is $G$-equivariant, we have $f \circ L_g = L_g \circ f$ for any $g \in G$. Also, since $\rho$ is $G$-invariant, we have $\rho \circ L_g = \rho$. Combining these properties, we see that:

$$\rho_f(L_g x) = \rho_f(L_g x) \frac{|\det J_{L_g}(x)|}{|\det J_{L_g}(x)|} = \frac{\rho_{R_{g^{-1}} \circ f}(x)}{|\det J_{L_g}(x)|} \qquad \text{(expanding definition of } \rho_f)$$

$$= \frac{\rho_{f \circ R_{g^{-1}}}(x)}{|\det J_{L_g}(x)|} = \rho\left((L_g \circ f^{-1})(x)\right) \frac{|\det J_{L_g \circ f^{-1}}(x)|}{|\det J_{L_g}(x)|} \qquad \text{(G-equivariance of } f)$$

$$= (\rho \circ L_g \circ f^{-1})(x) \frac{|\det J_{L_g}(f^{-1}(x)) J_{f^{-1}}(x)|}{|\det J_{L_g}(x)|} \qquad \text{(expanding Jacobian)}$$

$$= (\rho \circ f^{-1})(x) \frac{|\det J_{L_g}(f^{-1}(x))||\det J_{f^{-1}}(x)|}{|\det J_{L_g}(x)|} \qquad \text{(G-invariance of } \rho)$$

$$= \rho(f^{-1}(x))|\det J_{f^{-1}}(x)| \cdot \frac{|\det J_{L_g}(f^{-1}(x))|}{|\det J_{L_g}(x)|} \qquad \text{(rearrangement)}$$

$$= \rho_f(x) \cdot \frac{|\det J_{L_g}(f^{-1}(x))|}{|\det J_{L_g}(x)|} \qquad \text{(expanding definition of } \rho_f)$$

Now note that $G$ is contained in the isometry group, and thus $L_g$ is an isometry. This means $|\det J_{L_g}(x)| = 1$ for any $x \in \mathcal{M}$, so the right-hand side above is simply $\rho_f(x)$, which proves the theorem. $\qquad \square$

### A.4 Proof of Theorem 4

**Theorem 4.** *Let $(\mathcal{M}, h)$ be a closed Riemannian manifold. Let $\pi$ be a smooth, non-vanishing distribution over $\mathcal{M}$, which will act as our target distribution. Let $\rho_t$ be a distribution over said manifold parameterized by a real time variable $t$, with $\rho_0$ acting as the initial distribution. Let $D_{KL}(\rho_t \| \pi)$ denote the Kullback–Leibler divergence between distributions $\rho_t$ and $\pi$. If we choose a $g : \mathcal{M} \to \mathbb{R}$ such that*

$$g(x) = \log\left(\frac{\pi(x)}{\rho_t(x)}\right),$$

*and if $\rho_t$ evolves with $t$ as the distribution of a flow according to $g$, it follows that*

$$\frac{\partial}{\partial t} D_{KL}(\rho_t \| \pi) = -\int_{\mathcal{M}} \rho_t \exp(g) \|\nabla g\|^2 \, dx = -\int_{\mathcal{M}} \pi \|\nabla g\|^2 \, dx$$

*implying convergence of $\rho_t$ to $\pi$ in $KL$. Moreover, the exact diffeomorphism that takes us from $\rho_0 \to \pi$ is as follows. Given some initial point $x \in \mathcal{M}$, let $u(t)$ be the solution to the initial value problem given by:*

$$\frac{du(t)}{dt} = \nabla g(t), \qquad u(0) = x$$

*The desired diffeomorphism maps $x$ to $\lim_{t \to \infty} u(t)$.*

*Proof.* **1) Derivative of $D_{KL}(\rho_t \| \pi)$.** We start by noting the following: by the Fokker-Planck equation, $\rho_t$ evolving as a flow according to $g$ is equivalent to

$$\frac{\partial \rho_t}{\partial t} = \nabla \cdot (\rho_t \nabla g).$$

Please observe that since $\rho_t$ is defined as being a solution to the Fokker-Planck equation [39], $\rho_t$ will be a family of densities. In particular, the Fokker-Planck equation describes the time evolution of a probability density function.

Keeping Fokker-Planck in mind, we obtain the following expression for the time derivative of $D_{KL}(\rho_t||\pi)$:

$$\frac{\partial}{\partial t} D_{KL}(\rho_t||\pi) = \int \frac{\pi}{\rho_t} \frac{\partial \rho_t}{\partial t} \, dx$$

$$= \int \frac{\pi}{\rho_t} \nabla \cdot (\rho_t \nabla g) \, dx$$

$$= \int \left( \nabla \cdot \left( \frac{\pi}{\rho_t} (\rho_t \nabla g) \right) - (\rho_t \nabla g) \cdot \nabla \cdot \frac{\pi}{\rho_t} \right) dx$$

$$= \int \left( \nabla \cdot (\pi \nabla g) - (\rho_t \nabla g) \cdot \nabla \frac{\pi}{\rho_t} \right) dx$$

$$= - \int (\rho_t \nabla g) \cdot \nabla \frac{\pi}{\rho_t} \, dx,$$

where the final equality follows from the divergence theorem, since the integral of the divergence over a closed manifold is $0$. Now if we choose $g$ such that:

$$g(x) = \log \left( \frac{\pi(x)}{\rho_t(x)} \right).$$

Then we have:

$$\frac{\partial}{\partial t} D_{KL}(\rho_t||\pi) = - \int (\rho_t \nabla g) \cdot \nabla \exp(g) \, dx$$

$$= - \int \rho_t \exp(g) ||\nabla g||^2 \, dx,$$

**2) Proof of convergence.** Consider:

$$\frac{\partial \rho_t}{\partial t} = \nabla \cdot (\rho_t \nabla g)$$

where $g$ is defined as above. Note by standard existence and uniqueness results for differential equations on manifolds (for example, see do Carmo [11]) we have the existence of a solution, $\rho_t$ for all time $t > 0$, to this differential equation with initial value $\rho_0$.

Now note $g$, expressed as a function of $\rho_t$, is an invariant potential, the flow of which maps $\rho_0$ to $\lim_{t \to \infty} \rho_t$. By the result above, we know the right-hand-side of the equation:

$$\frac{\partial}{\partial t} D_{KL}(\rho_t||\pi) = - \int \rho_t \exp(g) ||\nabla g||^2 \, dx,$$

must approach $0$ (since the $KL$-divergence cannot continue decreasing at any constant rate, as it must be non-negative). The only way the right-hand-side can be $0$ is when $\nabla g = 0$, which can occur only when $\rho_t = \pi$. This concludes the proof of convergence of $\rho_t \to \pi$ in $KL$.

**3) Showing diffeomorphism $\rho_0 \to \pi$ is well-defined.** The exact diffeomorphism from $\rho \to \pi$ is as follows. Given some initial point $x \in \text{supp}(\rho)$, let $u(t)$ be the solution to the initial value problem given by:

$$\frac{du(t)}{dt} = \nabla g(t), \qquad u(0) = x$$

$g$ is defined as before. Note $u(t)$ exists and is unique by standard differential equation uniqueness and existence results [11]. We claim the desired diffeomorphism maps $x$ to $\lim_{t \to \infty} u(t)$. All that remains is to show (a) convergence to a smooth function at the limit and (b) that equivariance of the diffeomorphism does not break at the limit. We begin by showing this for $\pi$ uniform and finish the proof by extending to $\pi$ general.

$\pi$ **uniform.** For simplicity, we first consider the case where $\pi$ is the uniform (Haar) measure. In this case, the differential equation that $\rho$ obeys reduces to the heat equation, namely:

$$\frac{\partial \rho}{\partial t} = \Delta \rho$$

(a) Please note the following: an important fact that makes harmonic analysis on compact manifolds possible is that the spectrum of the Laplacian on any compact manifold must be discrete, i.e. its eigenvalues are countable and tend to infinity [12]. Also, its eigenvectors must be smooth (intuitively this says harmonic analysis is "nice" on manifolds in the same way that Fourier analysis is nice on the unit circle).

Note also that the Laplacian is Hermitian and negative semidefinite, and moreover that the only eigenfunction for eigenvalue $0$ is the constant vector.

Both facts above imply the solution to the above differential equation will just be the sum of several exponentially decaying (in $t$) terms and a constant term, given by the harmonic expansion of $\rho_0$.

From here, it follows that the $L^2$ distance between $\rho_t$ and the constant potential is just the sum of squares of the coefficients in front of those terms (this is simply the manifold analog of Parseval's theorem). However, all of those terms are decaying exponentially, so it follows that $\rho_t$ converges in the $L^2$ norm to the constant potential[7].

(b) Additionally, note that if the initialization $\rho_0$ is $G$-invariant, then it is fairly easy to see that all the terms in its harmonic expansion must also be $G$-invariant. As a result, $\rho$ must be $G$-invariant at all times, and must remain $G$-invariant in the limit. Similarly, its flow must be $G$-equivariant.

$\pi$ **general.** We have shown the desired properties for the case of $\pi$ uniform. However, the general case is entirely analogous, as the modified operator (involving $\pi$) has all the same relevant properties as the Laplacian (it is just generally better known that the Laplacian has these properties).

$\square$

### A.5 Conjugation by $SU(n)$ is an Isometry

We now prove a lemma that shows that the group action of conjugation by $SU(n)$ is an isometry subgroup. This implies that Theorems 1 through 3 above can be specialized to the setting of $SU(n)$.

**Lemma 2.** *Let $G$ be the group action of conjugation by $SU(n)$, and let each $L_g$ represent the corresponding action of conjugation by $g \in SU(n)$. Then $G$ is an isometry subgroup.*

*Proof.* We first show that the matrix conjugation action of $SU(n)$ is unitary. For $R, X \in SU(n)$, note that the action of conjugation is given by $\text{vec}(RXR^{-1}) = (R^{-T} \otimes R)\text{vec}(X)$. We have that $R^{-T} \otimes R$ is unitary because:

$$(R^{-T} \otimes R)^*(R^{-T} \otimes R)$$
$$= (\overline{R^{-1}} \otimes R^*)(R^{-T} \otimes R) \qquad \text{(conjugate transposes distribute over } \otimes\text{)}$$
$$= (\overline{R^{-1}R^{-T}}) \otimes (R^*R) \qquad \text{(mixed-product property of } \otimes\text{)}$$
$$= (R^T R^{-T}) \otimes (I) = (I) \otimes (I) = I_{n^2 \times n^2} \qquad \text{(simplification)}$$

Now choose an orthonormal frame $X_1, \ldots, X_m$ of $T\mathcal{M}$. Note that $T\mathcal{M}$ locally consists of $SU(n)$ shifts of the algebra, which itself consists of traceless skew-Hermitian matrices [19]. We show $G$ is an isometry subgroup by noting that when it acts on the frame, the resulting frame is orthonormal. Let $g \in G$, and consider the result of action of $g$ on the frame, namely $L_g X_1, \ldots, L_g X_m$. Then we have:

$$(L_g X_i)^*(L_g X_j) = X_i^* R_g^* L_g X_j = X_i^* X_j.$$

Note for $i \neq j$, we have $X_i^* X_j = 0$ and for $i = j$ we see $X_i^* X_i = 1$. Hence the resulting frame is orthonormal and $G$ is an isometry subgroup. $\square$

---

[7]Please note that if we wanted some other type of convergence, e.g. pointwise convergence, we could get this as well using a similar argument, by analyzing the decay properties of the eigenvalues/eigenvectors of the Laplacian.

# B   Manifold Details for the Special Unitary Group $SU(n)$

In this section, we give a basic introduction to the special unitary group $SU(n)$ and relevant properties.

**Definition.** The special unitary group $SU(n)$ consists of all $n$-by-$n$ unitary matrices $U$ (i.e. $U^*U = UU^* = 1$ for $U^*$ the conjugate transpose of $U$) that have determinant $\det(U) = 1$.

Note that $SU(n)$ is a smooth manifold; in particular, it has Lie structure [19]. Moreover, the tangent space at the identity (i.e. the Lie algebra) consists of traceless skew-Hermitian matrices [19]. The Riemannian metric is $\text{tr}(A^\top B)$.

## B.1   Haar Measure on $SU(n)$

**Haar Measure.** Haar measures are generic constructs of measures on topological groups $G$ that are invariant under group operation. For example, the Lie group $G = SU(n)$ has Haar measure $\mu_H : SU(n) \to \mathbb{R}$, which is defined as the unique measure such that for any $U \in SU(n)$, we have

$$\mu_H(VU) = \mu_H(UW) = \mu_H(U)$$

for all $V, W \in SU(n)$ and $\mu_H(G) = 1$.

A topological group $G$ together with its unique Haar measure defines a probability space on the group. This gives one natural way of defining probability distributions on the group, explaining its importance in our construction of probability distributions on Lie groups, specifically $SU(n)$.

To make the above Haar measure definition more concrete, we note from Bump [5, Proposition 18.4] that we can transform an integral over $SU(n)$ with respect to the Haar measure into integrating over the corresponding diagonal matrices under eigendecomposition:

$$\int_{SU(n)} f d\mu_H = \frac{1}{n!} \int_T f(\text{diag}(\lambda_1, \ldots, \lambda_n)) \prod_{i<j} |\lambda_i - \lambda_j| d\lambda.$$

Thus, we can think of the Haar measure as inducing the change of variables with volume element

$$\text{Haar}(x) = \prod_{i<j} |\lambda_i(x) - \lambda_j(x)|^2.$$

To sample uniformly from the Haar measure, we just need to ensure that we are sampling each $x \in SU(n)$ with probability proportional to $\text{Haar}(x)$.

**Sampling from the Haar Prior.** We use Algorithm 1 [32] for generating a sample uniformly from the Haar prior on $SU(n)$:

---
**Algorithm 1** Sampling from the Haar Prior on $SU(n)$

---
Sample $Z \in \mathbb{C}^{n \times n}$ where each entry $Z_{ij} = Z_{ij}^{(1)} + iZ_{ij}^{(2)}$ for independent random variables $Z_{ij}^{(1)}, Z_{ij}^{(2)} \sim \mathcal{N}(0, 1/2)$.
Let $Z = QR$ be the QR Factorization of $Z$.
Let $\Lambda = \text{diag}(\frac{R_{11}}{|R_{11}|}, \ldots, \frac{R_{nn}}{|R_{nn}|})$.
Output $Q' = Q\Lambda$ as distributed with Haar measure.

---

## B.2   Eigendecomposition on $SU(n)$

One main step in the invariant potential computation for $SU(n)$ is to derive formulas for the eigendecomposition of $U \in SU(n)$ as well as formulas for double differentiation through the eigendecomposition (recall that we must differentiate the $SU(n)$-invariant potential $\Phi$ to get $SU(n)$-equivariant vector field $\nabla\Phi$ and another time to produce gradients to optimize this). During the initial submission of our paper, a general implementation of this for complex matrices did not exist. Furthermore, while various specialized numerical techniques have been developed [41] to perform this computation, the implementation of these was unnecessary for our test cases of $n = 2, 3$. Instead, we derived explicit formulas for the eigenvalues based on finding roots of the characteristic polynomials (given by root

formulas for quadratic/cubic equations). Note that this procedure does not scale to higher dimensions since there does not exist a closed form solution for $n > 4$ [1]. However, concurrently released versions of PyTorch [36] introduced twice differentiable complex eigendecomposition, allowing one to easily extend our methods to higher dimensions.

### B.2.1 Explicit Formula for $SU(2)$

We now derive an explicit eigenvalue formula for the $U \in SU(2)$ case. Let us denote $U = \begin{bmatrix} a + bi & -c + di \\ c + di & a - bi \end{bmatrix}$ for $a, b, c, d \in \mathbb{R}$ such that $a^2 + b^2 + c^2 + d^2 = 1$ as an element of $SU(2)$; then the characteristic polynomial of this matrix is given by

$$\det(\lambda I - U) = (\lambda - (a+bi))(\lambda - (a-bi)) + (c+di)(c-di) = (a-\lambda)^2 + b^2 + c^2 + d^2 = \lambda^2 - 2a\lambda + 1$$

and thus its eigenvalues are given by

$$\lambda_1 = a + i\sqrt{1 - a^2} = a + i\sqrt{b^2 + c^2 + d^2}$$
$$\lambda_2 = a - i\sqrt{1 - a^2} = a - i\sqrt{b^2 + c^2 + d^2}$$

**Remark.** We note that there is a natural isomorphism $\phi : S^3 \to SU(2)$, given by

$$\phi(a, b, c, d) = \begin{bmatrix} a + bi & -c + di \\ c + di & a - bi \end{bmatrix}$$

We can exploit this isomorphism by learning a flow over $S^3$ with a regular manifold flow like NMODE [29] and mapping it to a flow over $SU(2)$. This is also an acceptable way to obtain stable density learning over $SU(2)$.

### B.2.2 Explicit Formula for $SU(3)$

We now derive an explicit eigenvalue formula for the $U \in SU(3)$ case. For the case of $U \in SU(3)$, we can compute the characteristic polynomial as

$$\det(\lambda I - U) = \det\left( \begin{bmatrix} \lambda - U_{11} & -U_{12} & -U_{13} \\ -U_{21} & \lambda - U_{22} & -U_{23} \\ -U_{31} & -U_{32} & \lambda - U_{33} \end{bmatrix} \right)$$
$$= \lambda^3 + c_2\lambda^2 + c_1\lambda + c_0$$

where

$$c_2 = -(U_{11} + U_{22} + U_{33})$$
$$c_1 = U_{11}U_{22} + U_{22}U_{33} + U_{33}U_{11} - U_{12}U_{21} - U_{23}U_{32} - U_{13}U_{31}$$
$$c_0 = -(U_{12}U_{23}U_{31} + U_{13}U_{21}U_{32} + U_{11}U_{22}U_{33} - U_{12}U_{21}U_{33} - U_{13}U_{31}U_{22} - U_{23}U_{32}U_{11})$$

Now to solve the equation

$$\lambda^3 + c_2\lambda^2 + c_1\lambda + c_0 = 0$$

we first transform it into a depressed cubic

$$t^3 + pt + q = 0$$

where we make the transformation

$$t = x + \frac{c_2}{3}$$
$$p = \frac{3c_1 - c_2^2}{3}$$
$$q = \frac{2c_2^3 - 9c_2c_1 + 27c_0}{27}$$

Now from Cardano's formula, we have the cubic roots of the depressed cubic given by

$$\lambda_{1,2,3} = \sqrt[3]{-\frac{q}{2} + \sqrt{\frac{q^2}{4} + \frac{p^3}{27}}} + \sqrt[3]{-\frac{q}{2} - \sqrt{\frac{q^2}{4} + \frac{p^3}{27}}}$$

where the two cubic roots in the above equation are picked such that they multiply to $-\frac{p}{3}$.

## C    Experimental Details for Learning Equivariant Flows on $SU(n)$

This section presents some additional details regarding the experiments that learn invariant densities on $SU(n)$ in Section 6.

For the evaluation, we found that ESS (effective sample size) was not a good metric to compare learned densities in this context. In particular, we noticed that several degenerate (mode collapsed) densities were able to attain near perfect ESS while completely failing on matching the target distribution geometry. Given that Boyda et al. [3] did not release code and reported ESS only for certain test cases, we decided to exclude ESS as a metric from our paper and instead relied directly on distribution geometry visualization.

### C.1    Training Details

Our DeepSet network [45] consists of a feature extractor and regressor. The feature extractor is a 1-layer tanh network with 32 hidden channels. We concatenate the time component to the sum component of the feature extractor before feeding the resulting 33 size tensor into a 1-layer tanh regressor network.

To train our flows, we minimize the KL divergence between our model distribution and the target distribution [34], as is done in Boyda et al. [3]. In a training iteration, we draw a batch of samples uniformly from $SU(n)$, map them through our flow, and compute the gradients with respect to the batch KL divergence between our model probabilities and the target density probabilities. We use the Adam stochastic optimizer for gradient-based optimization [23]. The graph shown in Figure 2 was trained for 300 iterations with a batch size of 8192 and weight decay setting of 0.01; the starting learning rate for Adam was 0.01, and a multi-step learning rate schedule that decreased the learning rate by a factor of 10 every 100 epochs was used. We use PyTorch to implement our models and run experiments [36]. Experiments are run on one CPU and/or GPU at a time, where we use one NVIDIA RTX 2080Ti GPU with 11 GB of GPU RAM.

We note that during our implementation, there are specific parts of the code that involved careful tuning for effective training. Specifically, we perturbed the results of certain functions and gradients by small constants to ensure numerical stability of the training process. We also spent some time tuning the learning rate and some ODE settings. More details can be found in the accompanying Github code.

### C.2    Conjugation-Invariant Target Distributions

Boyda et al. [3] defined a family of matrix-conjugation-invariant densities on $SU(n)$ as:

$$p_{toy}(U) = \frac{1}{Z} e^{\frac{\beta}{n} \operatorname{Re} \operatorname{tr}\left(\sum_k c_k U^k\right)},$$

which is parameterized by scalars $c_k$ and $\beta$. The normalizing constant $Z$ is chosen to ensure that $p_{toy}$ is a valid probability density with respect to the Haar measure.

More specifically, the experiments of Boyda et al. [3] focus on learning to sample from the distribution with the above density with three components, in the following form:

$$p_{toy}(U) = \frac{1}{Z} e^{\frac{\beta}{n} \operatorname{Re} \operatorname{tr}\left(c_1 U + c_2 U^2 + c_3 U^3\right)}$$

We tested on three instances of the density, also used in Boyda et al. [3]:

| set $i$ | $c_1$ | $c_2$ | $c_3$ | $\beta$ |
|---------|-------|-------|-------|---------|
| 1 | 0.98 | -0.63 | -0.21 | 9 |
| 2 | 0.17 | -0.65 | 1.22 | 9 |
| 3 | 1 | 0 | 0 | 9 |

Table 1: Sets of parameters $c_1, c_2, c_3$ and $\beta$ used in the $SU(2)$ and $SU(3)$ experiments

Note that the rows of Figure 2 correspond to coefficient sets $3, 2, 1$, given in order from top to bottom.

### C.2.1 Case for $SU(2)$

In the case of $n = 2$, we can represent the eigenvalues of a matrix $U \in SU(2)$ in the form $e^{i\theta}, e^{-i\theta}$ for some angle $\theta \in [0, \pi]$. We then have $\mathrm{tr}(U) = e^{i\theta} + e^{-i\theta} = 2\cos(\theta)$, so above density takes the form:

$$p_{toy}(U) = \frac{1}{Z} e^{c_1 \beta \cos\theta} \cdot e^{c_2 \beta \cos(2\theta)} \cdot e^{c_3 \beta \cos(3\theta)}.$$

### C.2.2 Case for $SU(3)$

In the case of $n = 3$, we can represent the eigenvalues of $U \in SU(3)$ in the form $e^{i\theta_1}, e^{i\theta_2}, e^{i(-\theta_1 - \theta_2)}$. Thus, we have

$$\mathrm{Re}\,\mathrm{tr}(U) = \frac{1}{3} \left( \cos(\theta_1) + \cos(\theta_2) + \cos(-\theta_1 - \theta_2) \right)$$

and thus

$$p_{toy}(U) = \frac{1}{Z} e^{\frac{c_1 \beta}{3} \left( \cos(\theta_1) + \cos(\theta_2) + \cos(-\theta_1 - \theta_2) \right)}$$
$$\cdot e^{\frac{c_2 \beta}{3} \left( \cos(2\theta_1) + \cos(2\theta_2) + \cos(-2\theta_1 - 2\theta_2) \right)}$$
$$\cdot e^{\frac{c_3 \beta}{3} \left( \cos(3\theta_1) + \cos(3\theta_2) + \cos(-3\theta_1 - 3\theta_2) \right)}$$

## D  Learning Continuous Normalizing Flows over Manifolds with Boundary

**Motivation.** Recall that learning a continuous normalizing flow over a manifold with boundary is not principled, and is rather numerically unstable, since probability mass can "flow out" on the boundary. In particular we noted in Section 1 that this was a major problem for the quotient manifold approach to learning invariant densities, since the quotient frequently has a nonempty boundary.

**Our Approach.** Our method enables learning flows over manifolds with boundary. One need only represent the manifold with boundary as a quotient of a larger manifold without boundary and learn with an invariant potential function that ensures the density descends smoothly from the larger manifold without boundary to the manifold with boundary.

**Example.** For instance, one can use our method to construct a flow over an interval. Notice that we can view an interval $I = [0, 1]$ as a manifold with boundary. The boundary consists of the two endpoints, $\{0, 1\}$. To use our method to learn a flow over this interval, we need only represent $[0, 1]$ as the quotient of $S^2$ by the isotropy group at the north pole, then apply the flow construction described in Section 5.2.1. The learned density assigns the same value to all points at the same latitude: clearly, this descends to a density over $[0, 1]$ by taking one representative point from each latitude circle. Notice that this works more generally: we can represent various manifolds with boundary as quotients of larger manifolds by isotropy groups. In particular, one can imagine using this method to replace neural spline flows [13], which carefully constructs noncontinuous normalizing flows over intervals.