# OpenReview forum: "Equivariant Manifold Flows"
_NeurIPS.cc/2021/Conference — NeurIPS 2021 Poster_

### Official Review · Reviewer_N2xA · 2021-07-02

**Rating:** 6
**Confidence:** 5

**Summary:**

This paper proposes an extension Continuous Manifold Normalizing Flows to Equivariant Manifolds where the the equivariance is with respect to an isometry group of the manifold. To this end, the paper lifts existing tools to a more manifold specific setting and empirically demonstrates the utility of the proposed approach on sampling from SU(N).  Overall, the paper is relatively well written and easy to follow. I found the main ideas, such as Theorems 1-3, interesting---albeit not extremely novel---and correct. I believe the empirical results, while a bit on the simpler end, are illuminating and see no reason to doubt the correctness of the numbers.

**Limitations And Societal Impact:**

The authors have adequately addressed their limitations and discussed negative societal impact.

**Main Review:**

First I'd like to highlight that the paper is quite easy to read and I had little trouble following the main results including the proofs in the Appendix. The main highlight of this paper for me were the two invariant potentials introduced which I believe are more of a central contribution than the developed theory.

This brings me the central complaint of the paper which is limited novelty. To be perfectly honest Theorems 1-3 are not sufficiently novel. In fact, I would not even elevate them to theorems but rather Lemmas as done in [1]. In fact [2] proves much of the results in Theorems 1-3 for group actions on Euclidean spaces. Lifting these to more general manifolds requires a little bit of technical nuance but I don't believe it is that difficult (e.g. instead of gradient, we use Riemannian gradient). Theorem 4 on the other seems to be more novel and interesting. I think it pokes at a larger but not fully explored issue on the universality of learning equivariant flows on manifolds. However, I believe this should be investigated further in the context of the proposed invariant potentials. A more powerful result would be to show that for a specific class of invariant potentials we can learn any equivariant density for some isometry group. Right if my understanding is correct the current result highlights that there exists an invariant potential where the KL is minimized and as a result this is a good learning target.

Finally, the experiments are also a bit light. It would've been interesting to propose new manifolds where existing approaches would definitely not work. Also, I would argue that there are missing baselines for the S^2 example. Finally, a lot of the results in the appendix are quite interesting and would've made for a stronger experimental section.

Overall, I believe this paper in its current narrative leans too heavily on borrowed theory which is extended in a natural but straightforward manner. Consequently, the main results that I found interesting such as the two proposed invariant potential functions and Theorem 4 are not given the full attention that they deserve. Consequently, I'm currently leaning towards a weak reject but I am willing to change my score if a good argument can be made against the points above and below.


Detailed Feedback:
------------------------------------------------------------------------------------------------------------------------------------------------------------------------------------
line 61 - Kohler et. al formulation for equivariant vector field is fairly general.
Section 3.2, this is technically correct but it would be more fitting and match the title if you decomposed the change of variable and explicitly show the role of the Riemannian metric

Theorem 1. Is a strict generalization from Kohler[2] & Papamakarios[1] review paper
Theorem 2-3 are direct consequences
R_g is a representation of the group. Is it a linear represenation? You must be more precise here.
- line 196 G-equivariant CNF, I can construct discrete NF's on manifold without this condition.
- you could also consider the prior density introduced in Kohler et. al which is on M-1 dimensional subspace.
- 5.2.2. you could also use uniform density here no?
- 359, Can you elaborate on what exactly you had to tune? This is an interesting point for the general community.


Point 2 line 45: We can always restrict the flow to a pre-defined support. This can be done by a projection operation. For example this is the underlying principle of a Restricted/Wrapped Gaussian. I don't believe this is an insurmountable limitation.

Point 3. Line 51. Can you expand more on this unnaturalness? Give a more precise characterization and explicate why exactly this is undesirable? It is possible to design flows over discrete structures in the conventional non-Manifold sense. Can we not leverage a similar design principle here? Please argue against this point.

Contribution 3 line 71. This is not really a new contribution but moreso the benefit of invoking symmetries when doing generative models. Other papers have also remarked the same observations. This paper also acknowledges this fact line 103-104


line 207. I disagree. It does not strictly reduce the problem as one can construct G-invariant densities but not going through a G-invariant potential function. This is a clear mistake. I believe what the authors intended to say that in this particular context it is sufficient to use G-invariant potential function. But indeed this is not necessary, for example I can construct an equivariant map that is not the gradient of an invariant potential function and still arrive at a G-invariant density.

Theorem 4. I believe I don't fully agree with the interpretation here. I agree that there may exist a G-invariant potential for every target distribution on a closed manifold. But you must show that the construction of one is universal. This point is missing.


For 6.2 The obvious missing baseline is NF's on Spheres and Tori in addition to the Manifold flow.

Minor Presentation issues:
Line 141, you haven't defined what a symmetry is.
Line 146: This is obvious, and here $f$ is not distinguished as a flow.
Overall 3.3 is a bit sloppy in notation and could be cleaned up a little bit.
X in line 174 is a vector field, earlier in the background it was a manifold.

Typo in line 506 in proof. I believe it should be D_u not Du R_g(x), ... in middle equality.

5.1 -> 5.2 This is confusing, first you assume that you're given an invariant potential function then immediately you design two new ones? From a narrative perspective this is not very clear.


Questions:
1. The prior for S^2 is only on the z-axis right ---i.e. its a 1-d uniform density?
2. 5.3 Isn't the conventional duality of Normalizing Flows. We either need to do density estimation or we need to generate high quality samples. I don't see how this is particularly unique to Manifold Flows.

------------------ Post-Rebuttal Updates -----------------------------------------------------------------------------------------------------------------------------------------
After fruitful discussion with the other reviewers and the detailed author comments to my questions, I have raised my score from 5->6. This is conditioned on the authors making significant changes to the writing as mentioned in my original review.

1.) The paper must do a better job of putting this work in context with [1] and [2] and clearly stating these are extensions of these ideas to the Riemannian case and not that they are proposed here for the first time. This is currently ambiguous in the original draft.

2.) Theorem 4 at submission time has many interesting nuances that were uncovered in the discussion. I suggest this theorem be re-written along with the proof which has a few technical issues that need to be addressed (e.g. smoothness in the limit) as pointed in the discussion.

I believe this paper has the potential to be very interesting to the community but the key detractor at the moment is the writing and more generally the unclear narrative in context with prior work. Once these issues are fixed---and if accepted---this paper would be interesting to the NeurIPS community.

[1] Papamakarios, G., Nalisnick, E., Rezende, D.J., Mohamed, S. and Lakshminarayanan, B., 2019. Normalizing flows for probabilistic modeling and inference. arXiv preprint arXiv:1912.02762.

[2] Köhler, J., Klein, L. and Noé, F., 2020, November. Equivariant flows: exact likelihood generative learning for symmetric densities. In International Conference on Machine Learning (pp. 5361-5370). PMLR.


**Time Spent Reviewing:**

8 hours

---

> ### Author Response · Authors · 2021-08-10
> **Response to Reviewer N2xA (1/2)**
>
> Thank you for your review and comments! We bold your comments and respond to them individually below. Due to the OpenReview character limit, we responded in two separate posts. This is post 1 out of 2.
>
> **This brings me the central complaint of the paper which is limited novelty. To be perfectly honest, Theorems 1-3 are not sufficiently novel. In fact, I would not even elevate them to theorems but rather Lemmas as done in [1].**
>
> While [2] proves rough analogues to our theorems 1-3 for Euclidean space, there are considerable differences. In addition to our statements implying stronger results on Euclidean space, we emphasize that our proof techniques are novel and interesting (they are geometric in nature while those in [2] are algebraic in nature).
>
> 1. Theorem 1 in [2] (rough Euclidean space analogue to our theorems 1 and 3) only proves results for matrix lie groups. They rely on the matrix structure to show their results, while our geometric proof uses the geometry of isometries. Note that our theorems imply results for E(n), which is the full isometry group and not a matrix lie group. This shortcoming in [2] is the basis for a moderately well known paper [4], meaning that this is substantial even for Euclidean space.
> 2. Theorem 2 in [2] (rough Euclidean space analogue to our theorem 2) relies on multiplication and addition commuting with integration. On general manifolds, such algebraic manipulations don’t exist; our geometric proof relies on the machinery of the fundamental theorem of flows and the naturality theorem of flows (lemma 1 in our paper is a corollary of both).
> 3. Since [2] relies so heavily on the niceties of the Euclidean metric and matrix algebra, their proof techniques do not generalize to a change in metric even with the same $\mathbb{R}^n$ underlying manifold.
>
> Given that the domain has been lifted so considerably, the symmetries are much more general, the Euclidean space analogues (which don’t even work on the full isometry group) were presented as theorems, the extra symmetries our results imply for Euclidean space equate to a paper [4], and the proofs require entirely different machinery, we believe that theorems 1-3 are novel and impactful.
>
> **(regarding Theorem 4) A more powerful result would be to show that for a specific class of invariant potentials we can learn any equivariant density for some isometry group.**
>
> Please consult the general comment that addresses this concern.
>
> **Also, I would argue that there are missing baselines for the S^2 example. (For 6.2 The obvious missing baseline is NF's on Spheres and Tori in addition to the Manifold flow.)**
>
> We do not include NF’s on Spheres and Tori as a baseline since they did not release code (reproducing is hard since certain implementation specific details may be omitted), and moreover the NMODE [3] experiments showed that NMODE is a strong baseline that outperforms NF’s on Spheres and Tori as well as other manifold flow methods. Hence we are using a very strong baseline and show that considerable improvement can be obtained with an equivariant $S^2$ flow.
>
> **Finally, a lot of the results in the appendix are quite interesting and would've made for a stronger experimental section.**
>
> For the initial submission, we were limited by space constraints and decided to move an experiment to the appendix, however we can move it back to the main paper with some reorganization. We appreciate that you think our experimental section is strong with the addition of the appendix results.
>
> [1] Papamakarios, G., Nalisnick, E., Rezende, D.J., Mohamed, S. and Lakshminarayanan, B., 2019. Normalizing flows for probabilistic modeling and inference. arXiv preprint arXiv:1912.02762.
>
> [2] Köhler, J., Klein, L. and Noé, F., 2020, November. Equivariant flows: exact likelihood generative learning for symmetric densities. In International Conference on Machine Learning (pp. 5361-5370). PMLR.
>
> [3] Aaron Lou, Derek Lim, Isay Katsman, Leo Huang, Qingxuan Jiang, Ser Nam Lim, and Christopher M De Sa. Neural manifold ordinary differential equations. In Advances in Neural Information Processing Systems, volume 33, pages 17548–17558, 2020.
>
> [4] Satorras, V. G., Hoogeboom, E., Fuchs, F. B., Posner, I., & Welling, M. (2021). E (n) Equivariant Normalizing Flows. arXiv preprint arXiv:2105.09016.

---

> ### Author Response · Authors · 2021-08-10
> **Response to Reviewer N2xA (2/2)**
>
> Thank you for your review and comments! We bold your comments and respond to them individually below. Due to the OpenReview character limit, we responded in two separate posts. This is post 2 out of 2.
>
> ### Response to detailed feedback, point-by-point:
>
> **line 61 - Kohler et. al formulation for an equivariant vector field is fairly general.**
>
> We stress that the Kohler et al. formulation is specific to Euclidean space, and moreover symmetry is with respect to action by subgroups of SO(n). This is less general than our formulation in two ways:
> 1. We allow for learning over more general manifolds (including more general Riemannian metrics).
> 2. We allow for learning with respect to more general symmetries (i.e. learning that is symmetric with respect to isometry subgroup action) while Kohler only allows for learning w.r.t. matrix Lie group actions on Euclidean space.
>
> **Section 3.2, this is technically correct but it would be more fitting and match the title if you decomposed the change of variable and explicitly show the role of the Riemannian metric**
>
> Section 3 is a brief background section, hence we tried to be as terse as possible. The Riemannian metric is inherently used when calculating the determinant (as this determines the volume of the basis). However, your concerns are valid, and we will expand upon this definition.
>
> **Theorem 1 is a strict generalization from Kohler[2] & Papamakarios[1] review paper Theorem 2-3 are direct consequences R_g is a representation of the group. Is it a linear representation? You must be more precise here.**
>
> Note that Theorem 1 is not just a strict generalization of the domain. In particular, Kohler et al. requires a matrix Lie group action (and thus constrains to a subgroup of SO(n)), while our theorem 1 shows the analogous result for all of the isometry group.
>
> Theorem 2 and 3 are not direct consequences as mentioned above. In particular, both theorems again are more general and theorem 2 is not direct since we can not rely on the niceties of integration on Euclidean space.
>
> $R_g$ is the application of g on the right side ($R_g(x) = x \cdot g$) and this notation is useful when taking derivatives.
>
> **line 196 G-equivariant CNF, I can construct discrete NF's on manifold without this condition.**
>
> We note that our wording of “G-equivariant flow” was referring to vector field flows in the traditional differential geometric sense. The point of Theorem 2 was to say that a vector field is equivariant if and only if the corresponding flow is. Indeed, there can be discrete NF’s on a manifold which are G-equivariant but do not arise from a vector field. We will clarify this in our updated version.
>
> **5.2.2. you could also use uniform density here no?**
>
> The Haar prior is the uniform density over this manifold that is Gauge equivariant in the desired sense (Boyda et al. uses the same prior density).
>
> **359, Can you elaborate on what exactly you had to tune? This is an interesting point for the general community.**
>
> We had to tune the learning rate, various limit constants for numerical stability, and ODE settings. We will be releasing code and all of our tuned settings will be publicly available. In addition, if we are accepted, we will append this discussion to our analysis (this was trimmed due to space concerns).
>
> **Point 2 line 45: We can always restrict the flow to a predefined support. This can be done by a projection operation. For example this is the underlying principle of a Restricted/Wrapped Gaussian. I don't believe this is an insurmountable limitation.**
>
> As mentioned, it is often difficult/impossible to principally project/restrict to a quotient manifold or any other submanifold.
> 1. For the restricted Gaussian example, we note that one has to normalize the remaining probability distribution left on the restricted set, which requires an exact evaluation of an integral of the probability distribution over the manifold. This is difficult and can only be done approximately.
> 2. For the wrapped Gaussian example, we note that it is often very difficult to evaluate probabilities for nontrivial topologies. For example, the wrapped Gaussian distribution on the circle requires an infinite sum to evaluate properly (which is why the von-Mises Fisher distribution is used in practice). Furthermore, wrapped distributions on the circle with non-sub-Gaussian tails are even more difficult to evaluate since tail decay is slow.
>
> **Point 3. Line 51. Can you expand more on this unnaturalness? Give a more precise characterization and explicate why exactly this is undesirable? It is possible to design flows over discrete structures in the conventional non-Manifold sense. Can we not leverage a similar design principle here?**
>
> We are unsure which “[flow] over discrete structures in the conventional non-Manifold sense” the review is referencing, but are willing to defend our analysis if a citation is provided. The flow over a discrete structure with which we are familiar is that provided in Boyda et al.; the paper effectively decomposes SU(n) into its eigenvalues and learns a flow over S(n) (the symmetric group). For SU(2) (as shown in our paper in figure 2a), we see this discrete flow has issues with minimizing mass near $\pm \pi$. For SU(3) (as shown in figure 2b) the edges in the modes are jagged, whereas ours are smooth due to the non-discrete construction of our flow.
>
> **Contribution 3 line 71. This is not really a new contribution but moreso the benefit of invoking symmetries when doing generative models. Other papers have also remarked on the same observations. This paper also acknowledges this fact line 103-104**
>
> We are the first to show equivariance is useful for manifold normalizing flows. We believe that this is a worthwhile contribution since:
> 1. Our construction of the equivariant vector field is completely different from the construction of general vector fields on manifolds given in [3]. It is important to show that the equivariant condition overcomes this potential limitation.
> 2. The fact that modelling isometry subgroup symmetry improves learning was not previously shown for equivariant flows over manifolds.
>
> **line 207. I disagree. It does not strictly reduce the problem as one can construct G-invariant densities but not going through a G-invariant potential function. This is a clear mistake. I believe what the authors intended to say that in this particular context it is sufficient to use G-invariant potential function. But indeed this is not necessary, for example I can construct an equivariant map that is not the gradient of an invariant potential function and still arrive at a G-invariant density.**
>
> We believe that the saying “x reduces to y” means that solving y is sufficient (but not necessarily required) to solve x. We believe that this is the standard nomenclature in fields such as theoretical computer science. In our context, we mean that it is possible to construct a G-invariant density if we can construct a G-invariant potential but it is not necessarily required to construct a G-invariant density through a G-invariant potential.
>
> **Theorem 4. I believe I don't fully agree with the interpretation here. I agree that there may exist a G-invariant potential for every target distribution on a closed manifold. But you must show that the construction of one is universal. This point is missing.**
>
> Please consult the general comment that addresses this concern.
>
> **For 6.2 The obvious missing baseline is NF's on Spheres and Tori in addition to the Manifold flow.**
>
> Please see the above comment about missing baselines. Briefly, we do not include NF’s on Spheres and Tori as a baseline since they did not release code (reproducing is hard since certain implementation specific details may be omitted in the paper), and moreover the NMODE [3] experiments showed that NMODE was a stronger baseline.
>
> **Minor Presentation issues: Line 141, you haven't defined what a symmetry is. Line 146: This is obvious, and here  is not distinguished as a flow. Overall 3.3 is a bit sloppy in notation and could be cleaned up a little bit. X in line 174 is a vector field, earlier in the background it was a manifold.**
>
> Thank you; we will update our final draft to correct these issues.
>
> **Typo in line 506 in proof. I believe it should be D_u not Du R_g(x), ... in middle equality.**
>
> Thank you; this will be corrected.
>
> **5.1 -> 5.2 This is confusing, first you assume that you're given an invariant potential function then immediately you design two new ones? From a narrative perspective this is not very clear.**
>
> Thank you; instead of “assume we are given G-invariant potentials $f$” we will say “we will show how to construct G-invariant potentials $f$ below; for now, assume a G-invariant potential f is given” to improve narrative clarity.
>
> **The prior for S^2 is only on the z-axis right ---i.e. its a 1-d uniform density?**
>
> It’s a 1-d density in the quotient space, but when we learn, we learn over the ambient space and hence the density prior is effectively uniform over the sphere.
>
> **5.3 Isn't the conventional duality of Normalizing Flows. We either need to do density estimation or we need to generate high quality samples. I don't see how this is particularly unique to Manifold Flows.**
>
> We re-emphasize the two paradigms for clarity.
>
> [1] Papamakarios, G., Nalisnick, E., Rezende, D.J., Mohamed, S. and Lakshminarayanan, B., 2019. Normalizing flows for probabilistic modeling and inference. arXiv preprint arXiv:1912.02762.
>
> [2] Köhler, J., Klein, L. and Noé, F., 2020, November. Equivariant flows: exact likelihood generative learning for symmetric densities. In International Conference on Machine Learning (pp. 5361-5370). PMLR.
>
> [3] Aaron Lou, Derek Lim, Isay Katsman, Leo Huang, Qingxuan Jiang, Ser Nam Lim, and Christopher M De Sa. Neural manifold ordinary differential equations. In Advances in Neural Information Processing Systems, volume 33, pages 17548–17558, 2020.

---

> ### Author Response · Authors · 2021-08-29
> **Thank you**
>
> We would like to thank the reviewer for participating in the discussion, which we believe has been fruitful. Given our author response and clarifications in the discussion, we respectfully ask if the reviewer believes this is enough to allay the original concerns and improve the given score?

---

> > ### Comment · Reviewer_N2xA · 2021-09-01
> > **Review Updated**
> >
> > Thank you for participating in the rebuttal and discussion of this paper. I have updated my review along with my score to reflect my current understanding of this work.

---

### Official Review · Reviewer_tx8j · 2021-07-09

**Rating:** 8
**Confidence:** 3

**Summary:**

The paper proposes a general method to construct equivariant continuous normalizing flows on Riemannian manifolds. For this, the authors built on (more pedestrian) results of Koehler et al using the fact that an invariant potential can be used to obtain an equivariant vector field. This vector field can then in turn be used to define a continuous normalizing flow which is equivariant under (a certain  left or right action of) the symmetry group. Finally, the push-forward of a invariant density under the equivariant flow is a invariant density with respect to the symmetry group.  Specifically, the manuscript considers the case for which G is (a subgroup of) the isometry group of the manifold.

The developed framework is then applied to SU(N) Lie groups for which the conjugation by SU(N) is (a subgroup of) the isometry group. This application is of relevance in the context of lattice field theory, i.e. numerical simulation of (mostly strongly interacting) quantum field theories, for which ML techniques have recently gained significant attention.

**Limitations And Societal Impact:**

Both limitations and societal impact are adequately discussed.

**Main Review:**

The paper is well-written, insightful and rigorous. The underlying theory is clearly explained (although the presentation necessarily requires a solid background in Differential Geometry). The proofs of the main results are derived in a transparent and pedagogical manner in the Supplement while the main text focuses on summarizing these main results and their interplay for the proposed methods which makes the paper very readable.

Personally, I find the experimental sections a bit weaker. The LQFT application is of high relevance but in this context one is mainly interested in gauge symmetries, i.e. each lattice site can be conjugated by different SU(N) group elements, while the corresponding density is invariant under such local symmetry actions. The authors mainly consider the case of a single lattice site. Of course, this is a very important building block for a fully gauge-invariant flow and thus still highly interesting. However, it would be preferable if the authors had demonstrated the superiority of their approach over existing results by Boyada et al by training a flow to approximate the density of e.g. 2d quenched SU(N) lattice gauge theory comparing for example the ESS or the error of the topological susceptibility for fixed number of samples. The reported results would suggest that their method also helps in this case but an experimental validation of this expectation would be desirable.


Apart from this weakness, I find the manuscript very convincing and insightful. I also want to encourage the authors to release their code such that the community can benefit from it.

A few minor suggestions:
L136-137: d and \partial an the equation should be used consistently in the numerator and denumerator
L189-190: larger spacing between the two equations would improve readability
L214, L216-218: distribution -> density
L506-507: Du -> D_u


========================== Update after rebuttal and discussions with authors/reviewers ==========================
I want to thank the authors and also the other reviewers for a very insightful discussion period. I want to strongly encourage the authors to implement the changes already suggested by the other reviewers.

**Time Spent Reviewing:**

3

---

> ### Author Response · Authors · 2021-08-10
> **Response to Reviewer tx8j**
>
> Thank you for your review and comments! We bold your comments and respond to them individually below.
>
> **However, it would be preferable if the authors had demonstrated the superiority of their approach over existing results by Boyda et al by training a flow to approximate the density of e.g. 2d quenched SU(N) lattice gauge theory…**
>
> We would have liked to do a comparison with Boyda et al. on a new application, but since Boyda et al. have not released code, there would be considerable practical issues in replicating their results and comparing on new test examples.
>
> **...comparing for example the ESS...**
>
> We found that ESS (effective sample size) was not a good metric to compare learned densities in this context. In particular, we noticed that several degenerate (mode collapsed) densities were able to attain near perfect ESS while completely failing target distribution geometry. Given that Boyda et al. did not release code and reported ESS only for certain test cases, we decided to exclude ESS as a metric from our paper and instead relied directly on distribution geometry visualization.
>
> **I also want to encourage the authors to release their code such that the community can benefit from it.**
>
> We plan on releasing code so that the community can benefit from our models.

---

### Official Review · Reviewer_RD2g · 2021-07-18

**Rating:** 7
**Confidence:** 5

**Summary:**

The paper investigates the construction of equivariant flows on Riemanian manifolds. It proposes to learn such flows by using continous flows, where the vector field is the gradient of an invariant function. Defining invariant functions is usually much easier than defining equivariant vector fields. This framework is then tested on S^2 with the action rotations along a fixed axis, and on SU(N) with the action of  SU(N) on itself by conjugation, for N=2,3.

**Limitations And Societal Impact:**

Continuous ODE flows are powerful but can be quite expensive, and ODE solvers do not usually deal with manifolds. The paper would benefit from having a short discussion on these points. For example, how many integration steps were used? Does this number vary over training? What ODE solver was used (Euler, some Runge-Kutta, ...)? What technique did the authors use to ensure their ODE solver stayed on SU(N) or S^2 (doing x + dt * grad_x is not going to stay on the manifold)?

**Main Review:**

Summary of the review: On the positive side, the approach of first building invariant functions in order to get equivariant vector fields is elegant. The experiments also show clear improvement over past work. On the negative side, I find the that authors over-claim their contribution, and Theorem 4 (which is crucial to prove that this approach is general enough) is badly written.


== Positive points ==
Moving the problems from "building equivariant maps to building invariant maps" is not just elegant, it is also of high practical value since building invariant functions is often much easier than building equivariant ones. This is particularly true when we also have the requirements that our functions should be invertible. The theory justifying this approach is laid out clearly, and the proof of Theorem 1 looks correct.

The experiments provided by the authors are convincing, and show clear improvements over the reference paper of Boyda et al.

== Negative points ==
A big over-claim I have found in the paper is the claim of "learning gauge invariant densities over SU(n) in the context of QFT". This claim appears in the abstract, and is repeated throughout the paper. This is totally incorrect. This papers addresses the issue of learning conjugation equivariant densities on SU(n), and tests it for n=2, 3. While learning such densities is a crucial step *towards* flows for QFT in Boyda et al, it is definitely not the whole story and this paper shows no QFT experiment. In short, the authors are right to use QFT as a motivation for their work, but they should *not* claim to be learning gauge invariant densities, or be doing QFT in this paper.

I tried really hard to understand Theorem 4 about the sufficency of flows generated by invariant functions, but failed to do so. The authors should rewrite this. For example (but this might not be the only problem), on line 219, how can then take a derivative with respect to time, when time is not used to define \rho or \pi?

In the experiment section, the authors build conjugation equivariant flows on SU(N) for N=2 or 3 only, when Boyda et al also showed results for SU(9). Did the authors find that there method was harder to implement for N > 3?


==========================
Update following rebuttal and discussions with authors/reviewers
==========================

The discussion with other reviewers and with the authors has been particularly fruitful. I am happy to raise my rating to "7. Good Paper - Accept", conditioned on the few points that were raised and discussed during the reviews and rebuttal period:
* make it clear (including in the abstract and introduction), that the paper does not contain any Lattice QCD or QFT experiment. Instead, lattice QFT is a motivation for learning SU(3) conjugation equivariant flows.
* as pointed out by other reviewers, the relationship of Theo 1-3 with Lemma 2 in [1] and Theo 2 in [2] should be made much clearer. In particular, Theo 1-3 are extensions to the Riemannian case of ideas that were previously explored in the Euclidean case.
* Theo 4 needs to be completely rewritten, using all the added explanations and improvements discussed during the rebuttal.

[1] Papamakarios, G., Nalisnick, E., Rezende, D.J., Mohamed, S. and Lakshminarayanan, B., 2019. Normalizing flows for probabilistic modeling and inference. arXiv preprint arXiv:1912.02762.

[2] Köhler, J., Klein, L. and Noé, F., 2020, November. Equivariant flows: exact likelihood generative learning for symmetric densities. In International Conference on Machine Learning (pp. 5361-5370). PMLR.

**Time Spent Reviewing:**

10

---

> ### Author Response · Authors · 2021-08-10
> **Response to Reviewer RD2g**
>
> Thank you for your review and comments! We bold your comments and respond to them individually below.
>
> **A big over-claim I have found in the paper is the claim of "learning gauge invariant densities over SU(n) in the context of QFT".**
>
> When we said that we "learned gauge invariant densities over SU(n) in the context of QFT", we meant to convey that our learned generative model is useful for the path integral computation in lattice QFT. However, as mentioned in the review, the path integral computation itself is outside of our paper’s scope. We understand that our wording could be misinterpreted and will revise it to be more clear.
>
> **I tried really hard to understand Theorem 4 about the sufficency of flows generated by invariant functions, but failed to do so.**
>
> Please consult the general reviewer comments for this concern.
>
> **In the experiment section, the authors build conjugation equivariant flows on SU(N) for N=2 or 3 only, when Boyda et al also showed results for SU(9). Did the authors find that there method was harder to implement for N > 3?**
>
> We built conjugation invariant flows only for SU(2) and SU(3) because they would be easiest to visualize and could better demonstrate the efficacy of our method. Furthermore, the SU(9) result in Boyda et al. was ancillary and was mainly used to show generality for SU(N).
>
> In theory, our method generalizes to arbitrary manifolds (in particular, SU(n) for all $n \in \mathbb{N}$). In practice, we were limited by PyTorch support for differentiation through complex eigendecomposition. In particular, our construction requires us to differentiate twice through eigendecomposition; this was not supported at the time we were writing this paper (which was in February of this year), hence we developed some numerical stability tricks unique to N=2, 3 (see appendices B.2.2 and B.2.3). That being said, very recently the PyTorch LinAlg library has come out with better support for complex autograd (the release we are referring to is V1.9 https://github.com/pytorch/pytorch/releases/tag/v1.9.0). Based on our early tests, we believe it should now be more viable to practically develop fully general SU(n) flows for n > 3.
>
> **Limitations**
>
> Most limitations that you have mentioned are not specific to our paper, but broadly to the neural manifold ODE literature. Many have been discussed in prior work [1], although we can certainly contribute to the mentioned aspects such as solver details and the manifold projection step. We shall add the specific details in a revision of the paper.
>
> [1] Aaron Lou, Derek Lim, Isay Katsman, Leo Huang, Qingxuan Jiang, Ser Nam Lim, and Christopher M De Sa. Neural manifold ordinary differential equations. In Advances in Neural Information Processing Systems, volume 33, pages 17548–17558, 2020.

---

> ### Author Response · Authors · 2021-08-29
> **Thank you**
>
> We would like to thank the reviewer for participating in the discussion, which we believe has been fruitful. Given our author response and clarifications in the discussion, we respectfully ask if the reviewer believes this is enough to allay the original concerns and improve the given score?

---

### Official Review · Reviewer_e1R7 · 2021-07-24

**Rating:** 5
**Confidence:** 1

**Summary:**

The paper develops a new method for learning densities over SU(n), and evaluates the performance for learning some idealized densities over SU(2) and SU(3), showing improvement over prior work of Boyda et al.


**Limitations And Societal Impact:**

Yes

**Main Review:**

Unfortunately the methods of this paper lie far outside my area of expertise so there is very little of substance I can offer in review.  However my guess is that, as currently written, it will be accessible and hence of interest only to a tiny fraction of the NeurIPS community.  This sentence for example seems to convey the main significance of the paper:
"Learning SU(n) gauge equivariant neural network flows is important for obtaining good flow-based samplers of densities on SU(n) useful for lattice quantum field theory [2]."
I have no idea what this means, and my guess is that many others will be in the same boat.  If the authors wish to reach more than just the theoretical physics/quantum field theory community, I would suggest that they put more work into writing this in a way that others can appreciate and learn something from that they can possibly apply to their own work.


**Time Spent Reviewing:**

1 hour

---

> ### Author Response · Authors · 2021-08-10
> **Response to Reviewer e1R7**
>
> Given that this is not your area of expertise, we would like to reiterate the relevance and purpose of our paper.
>
> Our work falls in the broader categories of machine learning on non-Euclidean spaces and equivariant machine learning. These areas are of particular interest to the NeurIPS community. Several papers have touched on these topics and have gained much traction in several recent machine learning conferences [3-5]. Additionally, NeurIPS 2020 hosted a workshop on machine learning methods for non-Euclidean spaces [1], and this year’s ICLR keynote was given on equivariant machine learning [2].
>
> We emphatically note that the quoted sentence does not highlight the main significance of our paper. It is instead a real life test case for which we apply our method.
>
> Rather, the purpose of our paper is to enable highly general density learning (by itself, density learning—learning to represent a probability distribution—is a fundamental problem in machine learning). Typically, density learning is done in Euclidean space (e.g. GANs, VAEs); geometric deep learning has recently aimed to extend these traditionally Euclidean methods to Riemannian manifolds, which are effectively the most general structures over which it makes sense to do calculus. Our paper presents density learning theory in this very general setting where not only does learning happen over Riemannian manifolds, but one can also learn with respect to specified symmetries.
>
> References:
>
> [1] NeurlPS 2020 Workshop on Differential Geometry meets Deep Learning,  https://sites.google.com/view/diffgeo4dl
>
> [2] Michael Bronstein, ICLR 2021 Keynote talk "Geometric Deep Learning: The Erlangen Programme of ML",  https://www.youtube.com/watch?v=w6Pw4MOzMuo
>
> [3] Aaron Lou, Derek Lim, Isay Katsman, Leo Huang, Qingxuan Jiang, Ser Nam Lim, and Christopher M De Sa. Neural manifold ordinary differential equations. In Advances in Neural Information Processing Systems, volume 33, pages 17548–17558, 2020.
>
> [4] Taco S. Cohen, Mario Geiger, Jonas Köhler, and Max Welling. Spherical CNNs. In International Conference on Learning Representations, 2018. Best paper ICLR 2018.
>
> [5] Nickel, M. and Kiela, D. Poincare embeddings for learning hierarchical representations. In Advances in Neural Information Processing Systems, pp. 6338–6347, 2017.

---

### Author Response · Authors · 2021-08-10
**General Reviewer Comments**

Thank you to all reviewers for the time they spent reviewing and their helpful comments. We have addressed reviewer-specific concerns in individual replies and look forward to further deliberation during the discussion period.

### Comments About Theorem 4

Several reviewers made comments about theorem 4. Our original intention with theorem 4 was to describe the flow between two smooth distributions, in the process proving existence of a flow that lets us learn to map (in particular) a $G$-invariant prior density to a $G$-invariant target density. Upon reading the reviews, we agree that our current statement may not adequately communicate this intent and plan on changing the statement of theorem 4 to the following:

$\textbf{Theorem.}$  Let $(\mathcal{M}, h)$ be a closed Riemannian manifold, and let $G$ be an isometry subgroup of $\mathcal{M}$. Let $\rho$ and $\pi$ be any smooth, nonvanishing distributions on $\mathcal{M}$ invariant to $G$. Then there exists a differentiable time-varying potential $g: \mathcal{M} \times [0,1] \rightarrow \mathbb{R}$ such that $g$ is invariant to $G$ and the flow generated by $g$ on the time interval $[0,1]$ maps $\rho$ to $\pi$.

For clarity, in the original Theorem 4 statement we let $\pi$ be a smooth, non-vanishing distribution over $\mathcal{M}$, which acted as our target distribution. We let $\rho$ be a distribution over $\mathcal{M}$ parameterized by a real time variable $t$. Here $g$ is parameterized by $t$ and one of $\rho$ or $\pi$ can be taken as the target.

Note that we can obtain the above result as a direct consequence of Theorem 4 (as currently stated) by reparameterizing the flow in time to flow over $[0, 1]$ instead of flowing over $[0, +\infty)$ and checking that continuity holds.

This change addresses the clarity concerns of reviewer RD2g by directly stating the theorem as the stronger result reviewer N2xA asked for.

---

> ### Comment · Reviewer_RD2g · 2021-08-12
> **Theorem 4**
>
> Thank you for clarifying Theorem 4. Elsewhere in the paper, $g$ is used to represent an element of the group. You should use $\Phi$ instead to denote a potential.
>
> Since the proof of Theorem 4 is still for the older statement, and since you justify this new version by saying it can be deduced from the old one, I still need to understand how to map from one version to the other.
>
> So, you're saying that in your original version, the density $\rho$ is dependent on $t$, so it's a family of densities (wrt the Haar measure) $\rho_t: M \rightarrow \mathbb{R}_+$. And that's where the dependence on $t$ comes from when taking the derivative of the KL wrt $t$. So far, this statement is very general, so how come you can end up with a derivative of the KL that is guaranteed to be negative? I could have chosen $\rho_t$ to move *away* from the target $\pi$. What assumption on $\rho$ am I missing here?

---

> > ### Author Response · Authors · 2021-08-14
> > **Theorem 4 Clarification**
> >
> > In the original version what we wished to communicate was that if we choose a $g: \mathcal{M} \rightarrow \mathbb{R}$ such that:
> >
> > $$
> > g(x) = \log \left( \frac{\pi(x)}{\rho(x)} \right)
> > $$
> >
> > then if $\rho$ evolves with $t$ as a flow according to $g$, it follows that:
> >
> > $$
> > \frac{\partial}{\partial t} D_{KL} (\rho || \pi) = - \int_{\mathcal{M}} \rho \exp(g) ||\nabla g||^2 dx
> > $$
> >
> > Notably $\rho$ evolving as a flow according to $g$ is equivalent by the Fokker-Planck equation to:
> >
> > $$
> > \frac{\partial \rho}{\partial t} = \nabla \cdot (\rho \nabla g)
> > $$
> >
> > from which we derived the result. Please note that $\rho_t$ is defined as a solution to a particular differential equation. We demonstrated the existence of a flow that takes us from $\rho$ to $\pi$ (flow dynamics specified by a particular $g$). This shall be made clear in the final version of the paper. In an effort to clarify your understanding, here is the original version of Theorem 4 with all clarifications in place:
> >
> > $\textbf{Theorem.}$ Let $(\mathcal{M}, h)$ be a closed Riemannian manifold. Let $\pi$ be a smooth, non-vanishing distribution over $\mathcal{M}$, which will act as our target distribution. Let $\rho$ be a distribution over said manifold parameterized by a real time variable $t$, and let $D_{KL}(\rho||\pi)$ denote the Kullback–Leibler divergence between distributions $\rho$ and $\pi$. If we choose a $g: \mathcal{M} \rightarrow \mathbb{R}$ such that
> >
> > $$
> > g(x) = \log \left( \frac{\pi(x)}{\rho(x)} \right),
> > $$
> >
> > then if $\rho$ evolves with $t$ as the distribution of a flow according to $g$, it follows that
> >
> > $$
> > \frac{\partial}{\partial t} D_{KL} (\rho \| \pi) = - \int \rho \exp(g) || \nabla g ||^2 dx
> > = - \int \pi || \nabla g ||^2 dx
> > $$
> >
> > where all integrals are taken over the whole manifold.
> >
> > **...how come you can end up with a derivative of the KL that is guaranteed to be negative?**
> >
> > Our argument is not that the KL-divergence must be sent to 0 (although this is in fact true), but rather that the right-hand-side of the equality must be sent to 0, since KL-divergence is non-negative. Note the right-hand-side only becomes 0 when the gradient of $g$ is 0, and this can only occur when $ \pi = \rho $.

---

> > > ### Comment · Reviewer_N2xA · 2021-08-17
> > > **Clarification**
> > >
> > > Hi,
> > >
> > > This restatement of Theorem 4 is more interesting. However, I cannot parse exactly where in the supplementary you have shown the existence of a flow that takes you $\rho \to \pi$. This is unfortunately not precise enough. Specifically, what is the exact equivariant diffeomorphism. Can you please clarify?

---

> > > > ### Author Response · Authors · 2021-08-17
> > > > **Clarification**
> > > >
> > > > Let us (1) recount the original argument, noting important details, and (2) clarify the desired equivariant diffeomorphism that takes us from $\rho \rightarrow \pi$.
> > > >
> > > > (1) Set $\rho_0$ to be the initial distribution we want to transform to $\pi$. Consider the differential equation:
> > > >
> > > > $$
> > > > \frac{\partial \rho}{\partial t} = \nabla \cdot (\rho \nabla g)
> > > > $$
> > > >
> > > > where $g$ is defined as in the theorem statement. Note by standard uniqueness and existence results for differential equations on manifolds (e.g. see do Carmo) we have the existence of a solution, $\rho_t$ for all time $t > 0$, to this differential equation with initial value $\rho_0$.
> > > >
> > > > Now note $g$, expressed as a function of $\rho_t$, is an invariant potential, the flow of which maps $\rho_0$ to $\lim_{t \rightarrow \infty} \rho_t$. By the result of Theorem 4, we know the right-hand-side of the equation:
> > > >
> > > > $$
> > > > \frac{\partial D_{KL} (\rho || \pi)}{\partial t} = - \int \rho \exp(g) || \nabla g ||^2 dx
> > > > $$
> > > >
> > > > must approach $0$ (the KL-divergence cannot continue decreasing at any constant rate, as it must be non-negative). The only way the right-hand-side can be $0$ is when $\nabla g = 0$, which can occur only when $\pi = \rho$. QED.
> > > >
> > > > (2) The exact diffeomorphism that takes us from $\rho \rightarrow \pi$ is as follows. Given some initial point $x$, let $u(t)$ be the solution to the initial value problem given by:
> > > >
> > > > $$
> > > > \frac{d u(t)}{dt} = \nabla g(t), u(0) = x
> > > > $$
> > > >
> > > > $g$ is defined as before. Note $u(t)$ exists and is unique by standard differential equation uniqueness and existence results. The desired diffeomorphism maps $x$ to $\lim_{t \rightarrow \infty} u(t)$. Alternatively, we can reparameterize this in time so that it happens over a finite time interval (this was the fashion in which we stated the new Theorem 4 in "General Reviewer Comments").

---

> > > > > ### Comment · Reviewer_N2xA · 2021-08-17
> > > > > **Thanks for the clarification**
> > > > >
> > > > > Thank you for providing this clarification. It is well noted. I am inclined to believe that this result is true but to be completely air tight it is necessary to show why we arrive at a smooth function at the limit. Correspondingly, it would be necessary to also show that in the limit equivariance (of the diffeo) /smoothness does not break. Are the authors able provide these details?

---

> > > > > > ### Author Response · Authors · 2021-08-17
> > > > > > **Further Details**
> > > > > >
> > > > > > We shall provide the requested details that make the argument "airtight", i.e. we show (1) convergence to a smooth function at the limit and (2) that equivariance of the diffeomorphism does not break at the limit.
> > > > > >
> > > > > > ### A. Case: $\pi$ uniform
> > > > > >
> > > > > > For simplicity, first please consider the case where $\pi$ is the uniform (Haar) measure. In this case, the differential equation that $\rho$ obeys reduces to the heat equation, namely:
> > > > > >
> > > > > > $$
> > > > > > \frac{\partial \rho}{\partial t} = \Delta \rho
> > > > > > $$
> > > > > >
> > > > > > (1) Please note the following: an important fact that makes harmonic analysis on compact manifold possible is that the spectrum of the Laplacian on any compact manifold must be discrete, i.e. its eigenvalues are countable and tend to infinity. Also, its eigenvectors must be smooth (intuitively this says harmonic analysis is "nice" on manifolds in the same way that Fourier analysis is nice on the unit circle).
> > > > > >
> > > > > > Note also that the Laplacian is Hermitian and negative semidefinite, and moreover that its only eigenvector is the constant vector.
> > > > > >
> > > > > > Both facts above imply the solution to the above differential equation will just be the sum of a bunch of exponentially decaying (in $t$) terms and a constant term, given by the harmonic expansion of $\rho_0$.
> > > > > >
> > > > > > From here, it follows that the $L^2$ distance between $\rho_t$ and the constant potential is just the sum of squares of the coefficients in front of those terms (this is simply the manifold analog of Parseval's theorem). However, all of those terms are decaying exponentially, so it follows that $\rho_t$ converges in the $L^2$ norm to the constant potential$^1$.
> > > > > >
> > > > > > (2) Additionally, note that if the initialization $\rho_0$ is $G$-invariant, then it is fairly easy to see that all the terms in its harmonic expansion must also be $G$-invariant. As a result, $\rho$ must be $G$-invariant at all times, and must remain $G$-invariant in the limit. Similarly, its flow must be $G$-equivariant.
> > > > > >
> > > > > > ### B. Case: $\pi$ general
> > > > > >
> > > > > > We have shown the desired properties for the case of $\pi$ uniform. However, the general case is entirely analogous, as the modified operator (involving $\pi$) has all the same relevant properties as the Laplacian (it's just generally better known that the Laplacian has these properties).
> > > > > >
> > > > > > #### Footnotes
> > > > > > $^1$Please note that if we wanted some other type of convergence, e.g. pointwise convergence, we could get this as well using a similar argument, by analyzing the decay properties of the eigenvalues/eigenvectors of the Laplacian.

---

> > > > > > > ### Comment · Reviewer_RD2g · 2021-08-20
> > > > > > > **Differential equation for distributions**
> > > > > > >
> > > > > > > Thanks a lot for all these clarifications.
> > > > > > >
> > > > > > > If I understand correctly, in Theo 4, $\rho_t$ is defined as the solution to the equation $\partial{\rho}{t} = \nabla\cdot(g\nabla g)$, where $g$ is a function of both $\pi$ and $\rho$. In particular you do no assume that $\rho$ is given by some other mean. Is that correctly?
> > > > > > >
> > > > > > > Assuming the above is correct, here is another question: knowing that there is a solution to the above equation is not enough, we also need to know that $\rho_t$ is a family of densities: they must remain non-negative, and their integral (using the Haar measure) must be $1$ for all $t$. I think we can prove the integral of $\rho_t$ is constant equal to $1$ using Stoke's theorem on a closed manifold (can you confirm?). How do you prove the function does not go negative at any point?

---

> > > > > > > > ### Author Response · Authors · 2021-08-21
> > > > > > > > **Fokker-Planck Equation**
> > > > > > > >
> > > > > > > > Your understanding is correct, assuming you made some minor typos in your comment. $\rho_t$ is the solution to:
> > > > > > > >
> > > > > > > > $$
> > > > > > > > \frac{\partial \rho}{\partial t} = \nabla \cdot (\rho \nabla g)
> > > > > > > > $$
> > > > > > > >
> > > > > > > > We assume only that the initial distribution $\rho_0$ is given.
> > > > > > > >
> > > > > > > > Please note that since $\rho_t$ is defined as the solution to the Fokker-Planck equation (https://en.wikipedia.org/wiki/Fokker%E2%80%93Planck_equation), $\rho_t$ will be a family of densities. **In particular, note that the Fokker-Planck equation describes the time evolution of a probability density function.**
> > > > > > > >
> > > > > > > > If you wish to confirm that the functional form of Fokker-Planck we use in the paper indeed corresponds to what you find in the Wikipedia entry, please consult the Wikipedia entry on the Convection-diffusion equation (https://en.wikipedia.org/wiki/Convection%E2%80%93diffusion_equation), which is functionally equivalent to Fokker-Planck.

---

### Decision · Program_Chairs · 2021-09-27

**Decision:**

Accept (Poster)

**Comment:**

The paper proposes a theoretical framework for constructing normalizing flows on Riemannian manifolds that are equivariant with respect to the action of an isometry group. This is an interesting contribution and highly relevant for distribution modelling under known symmetries in non-Euclidean spaces, which comes up often in problems in physics.

Overall, the reviews are positive so I'm happy to recommend acceptance. There has been a fruitful discussion between the reviewers and the authors that I hope will improve the final version further.

**Comments to the authors**: I would like to see the improvements suggested by the reviewers in the final version of the paper. For clarity, I'm copying below the changes requested by the reviewers that I'd like to see implemented:

1. Make it clear that the paper doesn't contain an application to QFT or Lattice QCD, but uses these as motivation only.
2. Moderate the novelty of Theorems 1-3 and make it clear how they relate to similar results from previous work.
3. Revise Theorem 4 and its proof following the discussion with the reviewers.

I would also like to highlight the review by Reviewer e1R7 and specifically this statement: "as currently written, [the paper] will be accessible and hence of interest only to a tiny fraction of the NeurIPS community". Indeed, I'd encourage you to reflect carefully on how this work can be communicated effectively so that it can maximize its future impact.